


# An analysis on temporal scaling behavior of extreme rainfall of Germany based on radar precipitation QPE data

Judith Marie Pöschmann[1], Dongkyun Kim[2], Rico Kronenberg[1], and Christian Bernhofer[1]

[1]Department of Hydrosciences, Institute of Hydrology and Meteorology, Technische Universität Dresden, 01069 Dresden, Germany

[2]Department of Civil and Environmental Engineering, Hongik University, Wausanro 94, Mapo-gu, Seoul, 04066, Korea

**Correspondence:** Judith Pöschmann (Judith.Poeschmann@tu-dresden.de)

**Abstract.** We investigate the depth–duration relationship of maximum rainfall over the whole of Germany based on 16 yrs of radar derived Quantitative Precipitation Estimates (namely, RADKLIM–YW, German Meteorological Service) with a space–time resolution of 1 km and 5 min. Contrary to the long–term historic records that identified a smooth power law scaling behavior between the maximum rainfall depth and duration, our analysis revealed three distinct scaling regimes of which boundaries are approximately 1 h and 1 d. Few extraordinary events dominate a wide range of durations and deviate to the usual power law. Furthermore, the shape of the depth–duration relationship varies with the sample size of randomly selected radar pixels. A smooth scaling behavior were identified when the sample size is small (e.g. 10 to 100), but the original three distinct scaling regimes became more apparent as the sample size increases (e.g. 1 000 to 10 000). Lastly, a pixel wise classification of the depth–duration relationship of the maximum rainfall at all individual pixels in Germany revealed three distinguishable types of scaling behavior, clearly determined by the temporal structure of the extreme rainfall events at a pixel. Thus, the relationship might change with longer time series and can be improved once available.

## 1 Introduction

Extreme rainfall poses significant threats to natural and anthropogenic systems (Papalexiou et al., 2016). The frequency and magnitude of extreme rainfall are expected to increase (Blanchet et al., 2016; Gado et al., 2017; García-Marín et al., 2012; Ghanmi et al., 2016; Lee et al., 2016; Madsen et al., 2009; Marra and Morin, 2015; Marra et al., 2016, 2017; Overeem et al., 2009; Yang et al., 2016) especially at sub–daily timescales (Fadhel et al., 2017; Westra et al., 2013, 2014) to which fatal and frequent disasters such as urban flash floods (Dao et al., 2020), riverine floods, and landslides sensitively react. Therefore, a thorough understanding on magnitude, duration, and frequency of extreme rainfall is imperative for efficient design, planning, and management of these systems. Probable maximum precipitation is one way to define extreme rainfall. It is defined as the theoretically greatest depth of precipitation for a given duration that is physically possible over a particular drainage basin at a particular time of year (Ralph E. Huschke, 1959). The PMP is primarily used while designing important hydraulic



structures such as major dams where breaking or failure would result in significant loss of lives and properties downstream. The PMP is estimated in three different ways. First, the hydro–meteorological method can be applied. In this method, the

maximum precipitation event that was observed at or nearby the location of interest is adjusted to account for the simultaneous occurrences of the most critical meteorological conditions (e.g. dew point, temperature, wind, and air pressures, etc.). Second, the statistical method fits the series of the annual maximum rainfall values to a given probability density function, and then the rainfall with extremely high recurrence interval (e.g. 10 000 yr) is estimated from the probability density function, which is considered as the PMP. Last, the maximum rainfall envelope curve method plots the depth (y)–duration (x) relationship of

the record rainfall events observed across a large geographical boundary (e.g. entire country or globe) on the log–log plane. Then, a straight line on the plot representing the upper boundary of the envelope containing all depth–duration relationships is estimated as the PMP. The maximum rainfall envelope curve method was first proposed by Jennings (1950), which showed that the depth of the extreme rainfall events observed across the globe is a power function of their duration. This relationship can be expressed as follows:

$$p = \alpha d^{\beta} \tag{1}$$

where p is the maximum precipitation occurring in duration d, the coefficient $\alpha$ represents the y intercept of the depth–duration relationship plotted on the log–log plane, and the exponent $\beta$ is the parameter characterizing the scaling behavior of the depth–duration relationship. Parameters $\alpha$ and $\beta$ have the value of 425 and 0.47 (for D in hours), respectively, for the analysis performed on the multiple extreme rainfall events observed across the globe. Jennings discovered that this unique scaling

behavior holds at the rainfall duration between 1 min through 24 months. Paulhus (1965) showed that the same power law relationship holds after addition of new world rainfall record observed at the island of La Réunion at the duration between 9 h and 8 d. Multiple exponents describing the scaling property of extremes have been retrieved at various regions around the world (Commonwealth of Australia, 2019; Gonzalez and Bech, 2017). Figure 1 shows the maximum rainfall–duration relationship identified by some of these studies. All relationships reveal power law relationships, with exponents around 0.45 (Spanish and

global estimate) and 0.2 (German) over a wide range of scales.

Several studies examined the validity of this universal scaling exponent. Galmarini et al. (2004) showed, based on the rainfall records observed at several stations in Canada, Australia, and La Réunion, that the single exponent scaling laws exist only for single stations experiencing extremely high precipitation and that the deviation from a scaling law is caused by the intermittency associated with a substantial number of zero precipitation intervals in data. They also showed that the scaling

exponent $\beta$ tend to stay around 0.5 based on the stochastic simulation assuming a point rainfall process composed of the Weibull distributed rainfall depth and a given temporal autocorrelation structure. Zhang et al. (2013) showed that the scaling exponent varies around 0.5, if the vertical moisture flux and rainfall can be modeled as the AR(1) and the truncated AR(1) stochastic process. However, these works showed the scaling behavior of maximum rainfall at a single point location, and did not investigate maxima observed at different spatial locations.

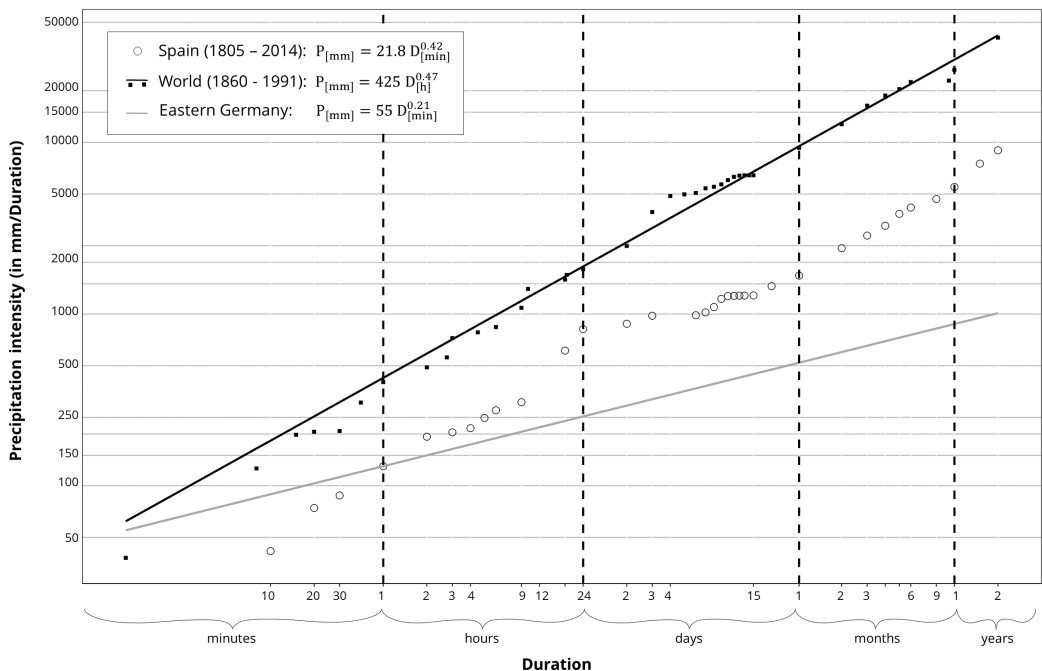

**Figure 1.** Extreme precipitation values for different durations: World extremes (World Meteorological Organization, 1994), Spanish extremes (Gonzalez and Bech, 2017), regional extremes for Germany (Dyck and Peschke, 1995)

One of the main obstacles to identify the "true" scaling behavior of maximum rainfall is that the most rainfall is measured from sparse ground gauge networks (Dyck and Peschke, 1995; Papalexiou et al., 2016). Remotely sensed precipitation products with high spatio–temporal resolution such as the ones provided by radar, satellite or microwave link networks may be applied to overcome this issue. Breña-naranjo et al. (2015) used a satellite based rainfall product to identify the scaling behavior of the maximum rainfall across the globe. They showed that the maximum of the areal rainfall averaged over the ~20 km × ~20 km data grid has the scaling exponent of ~0.43 which is similar to that of Jennings (1950) while the maximum rainfall values were systematically underestimated. They attributed the main reason of the discrepancy to the coarse spatial resolution of the satellite data that easily misses the small scale rainfall variability that is closely associated with extreme values (Cristiano et al., 2017; Fabry, 1996; Gires et al., 2014; Kim et al., 2019; Peleg et al., 2013, 2018).

Taken these findings, we want to analyze rainfall depth–duration relationship for the whole Germany based on the 16 y of the Quantitative Precipitation Estimates (QPE) radar product with 1 km–5 min space–time resolution in order to answer the following questions regarding the scaling behavior of the maximum rainfall: (1) Does the depth–duration relationship of German extreme rainfall show scale invariant behavior? If so or if not, what is the primary reason? (2) Does this relationship vary with regard to the spatial sampling rate? (3) Does it provide any clue to modify the relationship currently applied in practice based on sparse gauge network? The answers to these questions would be especially intriguing because few studies





have so far investigated the scaling behavior of maximum rainfall based on such a high spatio–temporal resolution rainfall data recorded over a long period and a large spatial extent as this study did. In addition, the practicality of the result would add value to this study. This is especially because the depth–duration relationship of the extreme rainfall of Germany revealed by our extensive QPE radar data analysis may be used to modify the current design guidelines regarding the PMP.

## 2   Data and Methods

### 2.1   Data Description

#### 2.1.1   Quantitative Precipitation Estimates (QPE)

The German National Meteorological Service (DWD) has been running a radar network (currently 18 C–band radars) for almost two decades and is providing different free and purchasable rainfall data derived from it. Full coverage of Germany has not been reached until today, however, all neighboring countries contribute to the rainfall information and the extension of
the network is ongoing. One QPE from German radar data is a Radar Online Calibration called RADOLAN (German: RADar OnLine ANeichung), which combines ground near information of fallen precipitation (rain gauge data) with radar data. The RADOLAN product is going through quality enhancements since 2005. A so–called radar climatology project of the DWD, RADolanKLIMatologie (RADKLIM, Winterrath et al., 2017) has consistently reanalyzed the complete radar data set since 2001 to attain homogeneity of the data that were processed through different algorithms. The RADKLIM data is available
in the following two formats: 1) RADKLIM–RW is an hourly precipitation product resulting from radar based precipitation estimates that are calibrated with ground stations (Winterrath et al., 2018a), which was validated by several studies, such as Lengfeld et al. (2019) and 2) RADKLIM–YW (Winterrath et al., 2018b) is a 5 min product resulting from a correction/factoring of DWD's 5 min product RADOLAN–RY with the help of RADOLAN–RH and RADKLIM–RW on a sequential hourly base. The RADKLIM–YW version 2017.002 was used in this study due to its high temporal resolution necessary for the analysis. It
is already the third version, covering the years 2001 to 2018. Due to comparison reasons with another study at our institute, only years 2001 to 2016 had been used for this study. The YW product covers the area composed of 1100 x 900 pixels with the spatial resolution of 1 km (improved compared to former version of RADOLAN). For more information on the RADKLIM–YW, see Kreklow et al. (2019).

#### 2.1.2   Data processing and quality issues

The data comprises 75.2 GB of compressed raw binary data (one layer for each time step) for 2001 to 2016, making 3.43 TB of unpacked data. Since not all raster cells are with values (only around half of the values lay within the borders of Germany), the spatial data was converted to time series for quicker processing. Analysis was conducted in R (R Core Team, 2019) and Matlab, with figures produced using ggplot (Wickham, 2016) and RasterVis (Perpiñán and Hijmans, 2019). The fst format (Klik, 2019) was chosen to store the data. The data contains the missing values (NA) of the following two types: 1) NAs due to changes
and ongoing extension of the radar network. This mainly affects areas near the border of Eastern, Northern, and Southern

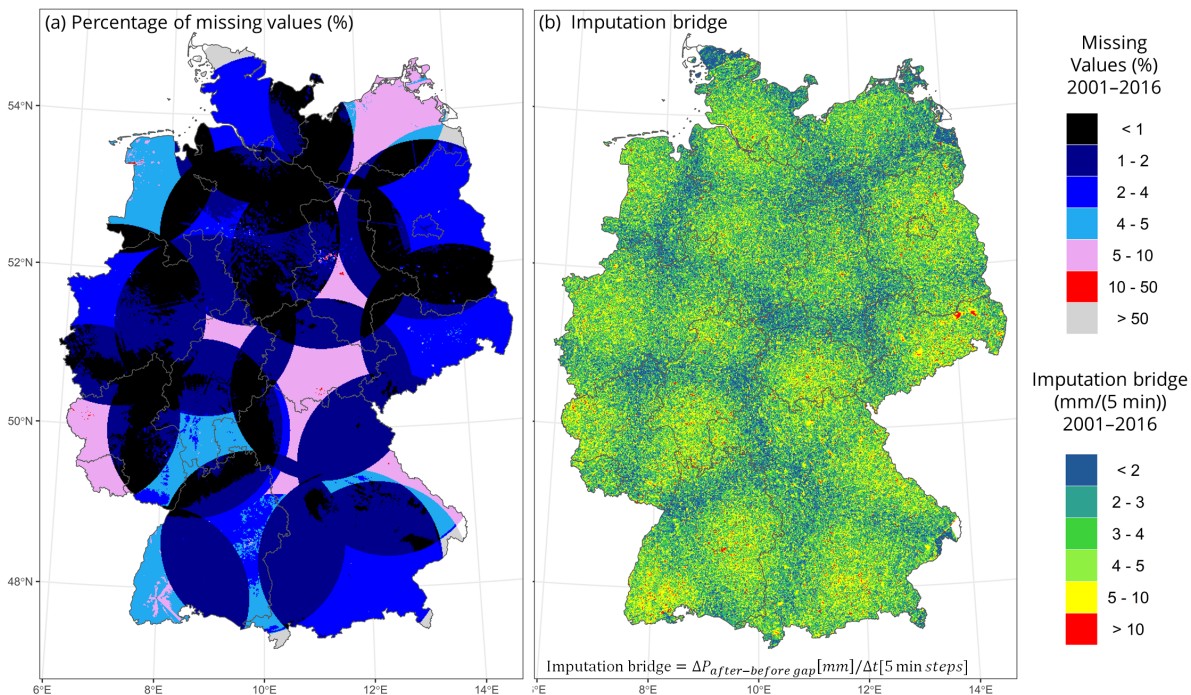

**Figure 2.** Spatial distribution of the proportion of missing values for the QPE RADKLIM–YW from 2001–2016: (a) Percentage of missing values (in %), (b) Maximum intensity per time step of 5 min that need to be interpolated = maximum intensity difference within one time step to overcome)

Germany. Some time series are, for example, only available from 2014 onwards. 2) Some locations/raster cells have NA values potentially due to malfunction of the radar or general (radar) errors. Figure 2 (a) shows the proportion of the NA values of the time series developed for each of the pixels. The visible cones display the individual radar coverage, and the overlapping areas of the radar cones have a better data coverage than the areas without overlapping. Additionally, Fig. 2 (b) shows the maximum
rainfall differences between right before and after a data gap, calculated for a time step of 5 min (Imputation bridge = Intensity difference/gap length). The red spots could mean a difference of greater than 180 mm h$^{-1}$.

There are multiple options of handling missing values, but it is very hard to handle highly episodic geophysical events such as rainfall. Even harder is the imputation of NA values which might be potential extreme rainfall events. We chose not to do any correction, since the distribution of the imputation bridge is sporadic, which means that the maximum values that
could have been missed at one pixel would be complemented by the observation at the adjacent pixel. Furthermore, the worst consequence of this approach would mean the missing of an extreme value whereas the consequences of imputing potentially too high extreme values seemed more severe and uncertain.





## 2.2 Methodology

### 2.2.1 Estimation of maximum rainfall at each radar cell

Maximum rainfall intensities were retrieved from aggregating the 5 min values for each duration $\tau$ of interest with moving window sampling using the R package RcppRoll (Ushey, 2018). Durations of up to 3 h were chosen for the analysis, with multiple steps for min and h out of our interest for sub–hourly and sub–daily pattern. The values were not aggregated spatially, since this usually reduces the maximum intensity values (Cristiano et al., 2018).

The radar data can be expressed as

$$X = \left\{ \{X_1,...,X_{n_P}\}_1, \{X_1,...,X_{n_P}\}_2,...\{X_1,...,X_{n_P}\}_N \right\} \tag{2}$$

with N being the number of pixels for whole Germany and $n_p$ the number of observation in time series.

For one single time series/raster cell the values $\left\{X_1,...,X_{n_P}\right\}_{cell}$ are aggregated for each $\tau$ according to Eq. (3), p being the index for each aggregated sample:

$$M_{\tau,p,cell} = \sum_{i=p}^{p+\tau-1} \{X_i\}_{cell}, p = 1,2,..(n_P - \tau + 1) \tag{3}$$

The maxima for each $\tau$ are retrieved with Eq. (4).

$$M_{max,cell}^{(\tau)} = \max \left\{ M_1^{(\tau)},.., M_{(n_P-\tau+1)\rfloor}^{(\tau)} \right\}_{cell} \tag{4}$$

### 2.2.2 Depth–Duration relationship for whole Germany

Finally, the overall maxima for each $\tau$ are taken from each time series' maxima according to Eq. (5).

$$M_{max}^{(\tau)} = \max \left\{ M_{max,1}^{(\tau)},.., M_{max,N}^{(\tau)} \right\} \tag{5}$$

The scaling relationship with scaling factor b and intercept B can be established as following.

$$\log(M_{max}^{(\tau)} = B + b \cdot \tau \tag{6}$$

respectively

$$M_{max}^{(\tau)} = 10^B \cdot b \tag{7}$$



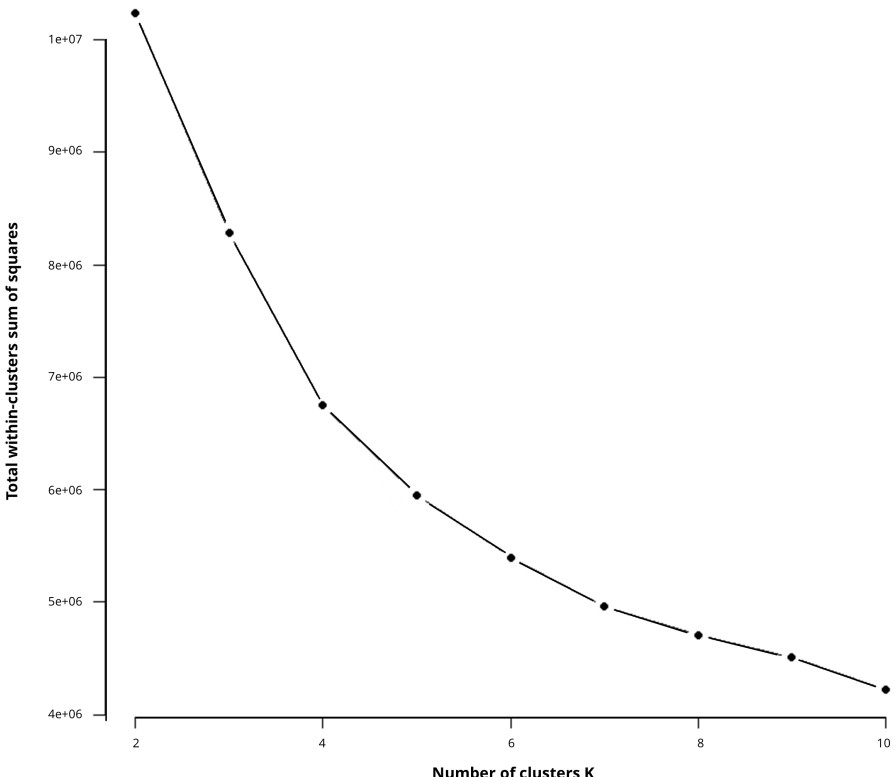

**Figure 3.** Elbow evaluation for K–Mean Clustering carried out for the rescaled maximum rainfall values

### 2.2.3 K–Mean Clustering of Depth-Duration-Relationships

The depth–duration relationships for each pixel derived from Sect. 2.2.1 are individually clustered with the K–Mean clustering algorithm (Scott and Knott, 1974). Erroneous pixels (due to too many NA values in the time series) were excluded from the cluster process in order to avoid disturbance. The data was rescaled to make the characteristics more comparable with each other. The elbow evaluation (compare Fig. 3 for estimating the best number of clusters gave no obvious suggestion (curve should bend like an elbow). We chose six clusters for a sufficiently detailed analysis.

## 3 Results and discussion


### 3.1 Scaling behavior of entire Germany

Figure 4 shows the maximum depth–duration relationship of the entire Germany that was derived from the QPE radar data (blue solid line). The same relationship based on ground gauge network (red dotted line) and the world record are shown for reference. The rain gauge based values clearly follow a scaling relationship with a slope that is different in comparison to





world extremes. Radar based maxima for the shorter duration from 2001 to 2016 do not cover all sub–daily extremes but exceed
observed ones from the 1 d durations as well as for some sub–daily values. In Fig. 4, a "plateau" is visible between around 35
min up to 18 h, indicating a "one event" effect at 35 min, potentially from an extreme rainfall event in this period. Overall, a
scaling behavior can be observed at sub–hourly durations with a scaling component of around 0.65 even though the maxima
were observed at very distant places of Germany as indicated by the map showing the location of the maximum rainfall. This

result implies that even though the location of extreme rainfall is different, the maximum rainfall may exhibit smooth scaling
behavior if the rainfall generation mechanism is similar. Between 25 min and 16 h, maximum values are calculated for the
southeastern edge of Hesse state in May 29th 2016. These very high values at this specific location are not documented in
public news, however there had been severe weather around that time. The extreme event around June 29th, 2017 at Berlin
comprised the maximum depth–duration relationship at the duration between 18 h and 2 d. The weak scaling behavior existed

in the regime at the 18 h and 3 d duration with the scaling exponent of 0.20.

### 3.2    Scaling behavior of entire Germany for high–quantile rainfall

High rainfall values are associated with especially great uncertainty when obtained from radar data. Thus, we also investigated
the scaling behavior of high–quantile rainfall values. Figure 5 shows the maximum depth–duration relationship of several
quantiles: 0.99999 (fourth greatest cells), 0.9999 (39th greatest cells), 0.999 (392nd greatest cells), and 0.99 (3921st cells). The

"three phase regime" from radar maximum values remains relatively stable, however, the "single event" effect between 50 min
and 1 d is smoothed out, because the degree of inflections in the curve becomes weaker. The lower the chosen quantile, the
clearer the scaling regime appears.

Figure 6 shows the location of the 0.9999, 0.9999, 0.999, and 0.99 quantile rainfall. The color of the circles represents the
different rainfall durations. The lower the quantile, the sparser the location of the quantile rainfall occurrence, which suggests

the reduction of the influence of one single rainfall event on the depth–duration relationship causing inflection in the curve.

The map corresponding to the lower quantiles (e.g. Fig. 6 (c) and 6 (d)) reveals that the locations of maximum rainfall
contributing to the development the rainfall–duration relationship are spread over the whole of Germany. Naturally, one would
assume that this heterogeneity of the meteorological conditions and rainfall generating mechanisms will reflect rather regional
characteristics and will exhibit some irregular scaling behavior. Contrary to this conjecture, the curve shows a very smooth

scaling behavior, that suggests the extreme rainfall events at this degree of quantile (upper one percentile) share common
characteristics such as peak rainfall depth and correlation structures regardless of timescale that could have been derived from
similar generation mechanisms.

### 3.3    Spatial distribution of maximum rainfall

Figure 7 shows the spatial distribution of 5 min, 30 min, 1 h, 6 h, 1 d, and 3 d maximum rainfall over Germany. The red and

yellow spots that are spatially distributed in Fig. 7 (a) suggest that 5 min extreme rainfall can happen at any place in Germany.
Note that extreme rain occurred also outside the Alpine region at the southern edge of Germany, which suggests that fine–scale
extreme rainfall is not necessarily governed by topography. The influence of fine–scale intense rainfall persists until hourly



**Figure 4.** Maximum depth–duration relationships and locations of rainfall maxima. Lines: German radar derived data of this study and German ground network (DWA, 2015; DWD, 2002, 2016), Spanish ground gauge records (Gonzalez and Bech, 2017), and world records (World Meteorological Organization, 1994). Map: Locations of rainfall maxima (based on the German radar data) for the considered duration.

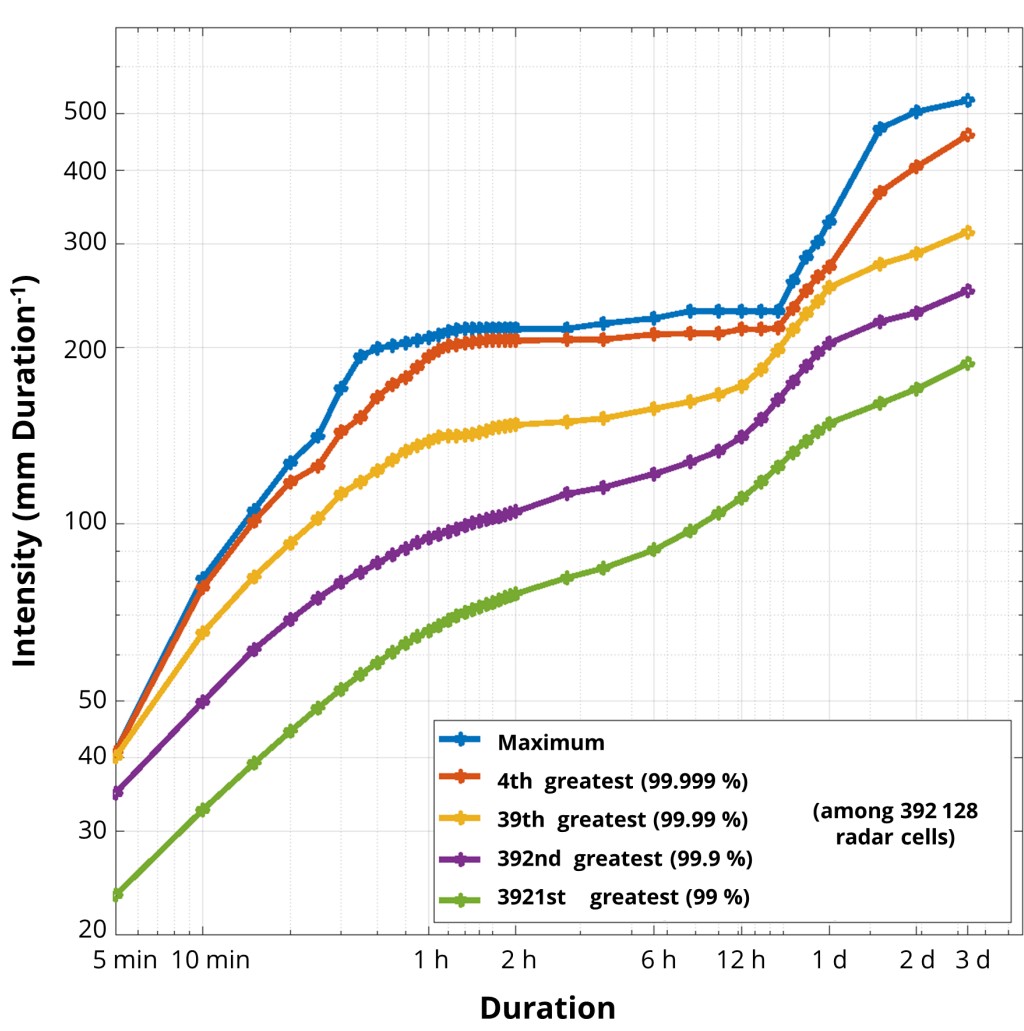

**Figure 5.** Depth–duration relationship of the maximum rainfall values of whole Germany down to the 3921st greatest per duration.
# Rainfall Maxima

**Figure 6.** Locations of the 0.99999, 0.9999, 0.999, and 0.99 quantile rainfall with varying durations from 5 min to 3 d. Point colors represent the corresponding rainfall duration, similar for each quantile.
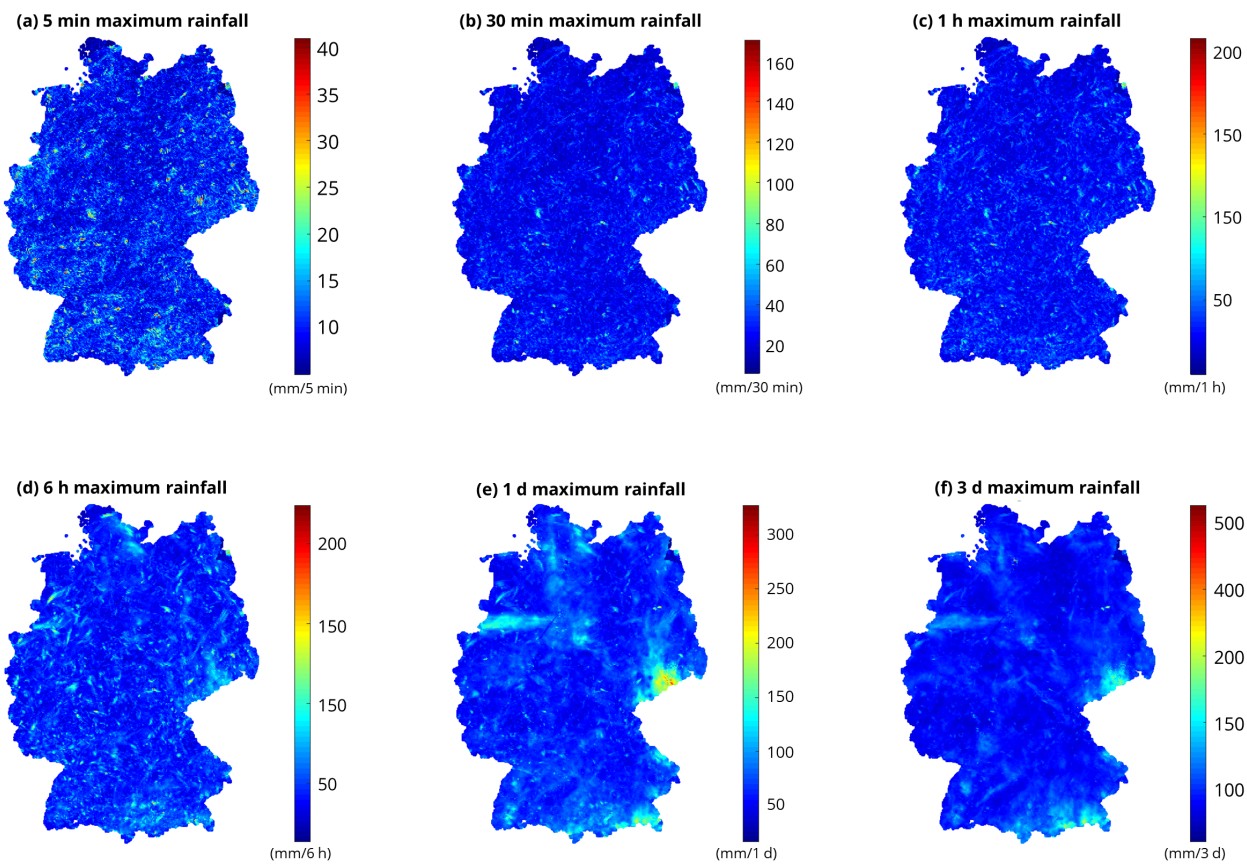

**Figure 7.** Spatial distribution of the maximum rainfall with a given duration over Germany

timescale as implied by the red and yellow hotspots that are similarly located in the maps of 5 min, 30 min, and 1 h. The distribution of maxima significantly changes for the 6 h duration (Fig. 7(d)), and an interesting pattern emerges in the map of 1 d and 3 d duration. These maxima seem to be dominated by single events/single rainfall that has been really high. Especially the 2002 flooding in Saxony (mid eastern edge) with precedent longer heavy rainfall as well as one singular rainfall event in 2014 (narrow aisle in the Northwestern area) are clearly visible in the maps.

## 3.4 Scaling behavior at a single point

Figure 8 shows the maximum rainfall–duration relationship of the radar rain cells at the major cities of Germany. Except for Hamburg, most cities exhibit slight (Hannover, Kiel, Magdeburg, Schwerin, Stuttgart) to considerable (the remaining cities) deviation from a single power law behavior. This significant deviation is similar to what was identified by Galmarni et al. (2004) which found that the inflection of the curve is inevitable because of the small (or zero) rainfall observations attached to the maximum rainfall event. This result also implies the real rainfall process significantly deviates from the assumptions of the

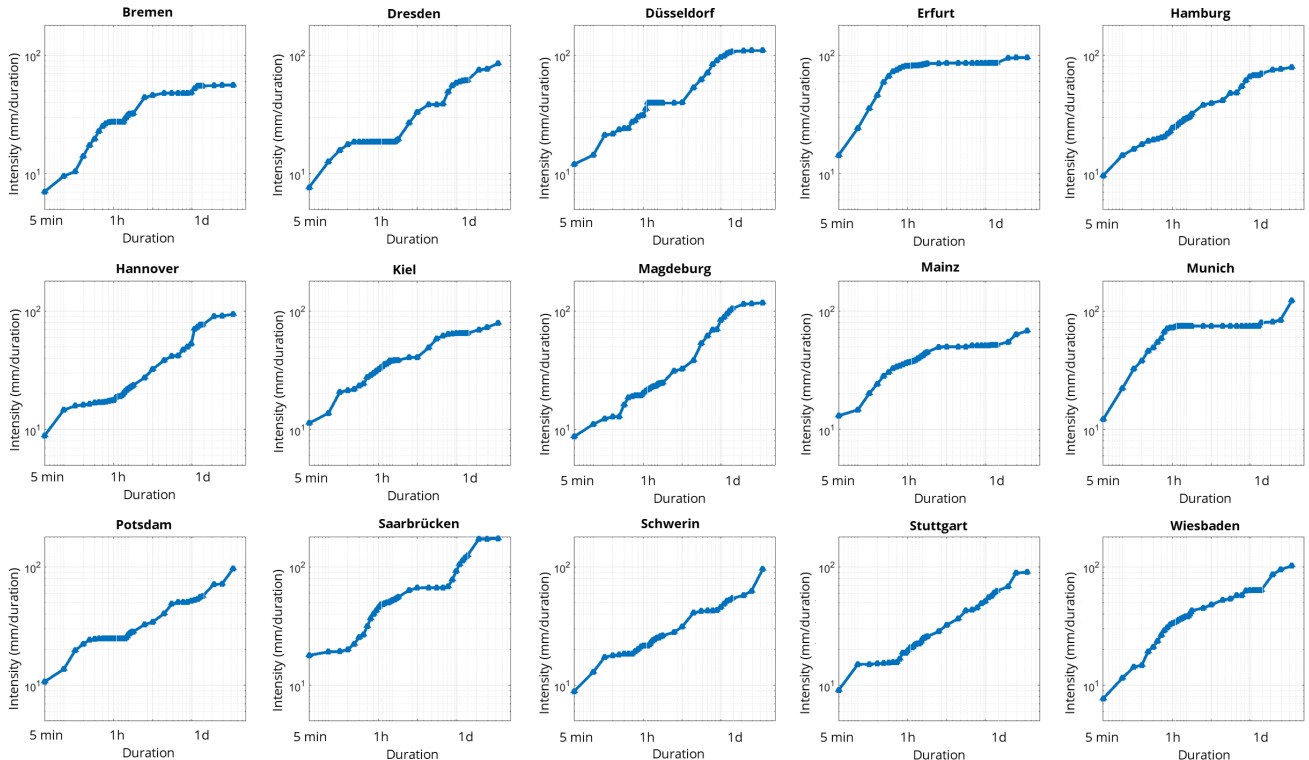

**Figure 8.** Maximum depth–duration plot of major cities of Germany

simple rainfall models suggested by Galmarini et al. (2004) and Zhang et al. (2013). Both studies showed that the maximum
rainfall–duration relationship at a given point location follow smooth and simple power law if the rainfall process can be
modeled with a set of simple stochastic processes.

### 3.5   Classification of maximum depth–duration relationship

The maximum dept–duration relationships in Fig. 8 were clustered since some show a similar shape with each other . The
k–mean clustering algorithm successfully classified the depth–duration relationship into six categories revealing different curve
characteristics regarding the curve shapes. Figure 9 shows a categorical map of Germany indicating each category with a certain
color and the depth–duration relationship at 100 grid cells randomly sampled from each category.

Cells belonging to Category 1 have the highest rainfall intensities over all scales until 1 d and show a strong inflection
at around 1 h similar to the scaling curve for whole Germany (Fig. 4). The behavior of the curve between 5 min and 1 h
is associated with strong convective rainfall events that pour for around 1 h and move on or weaken. Thus, these events are

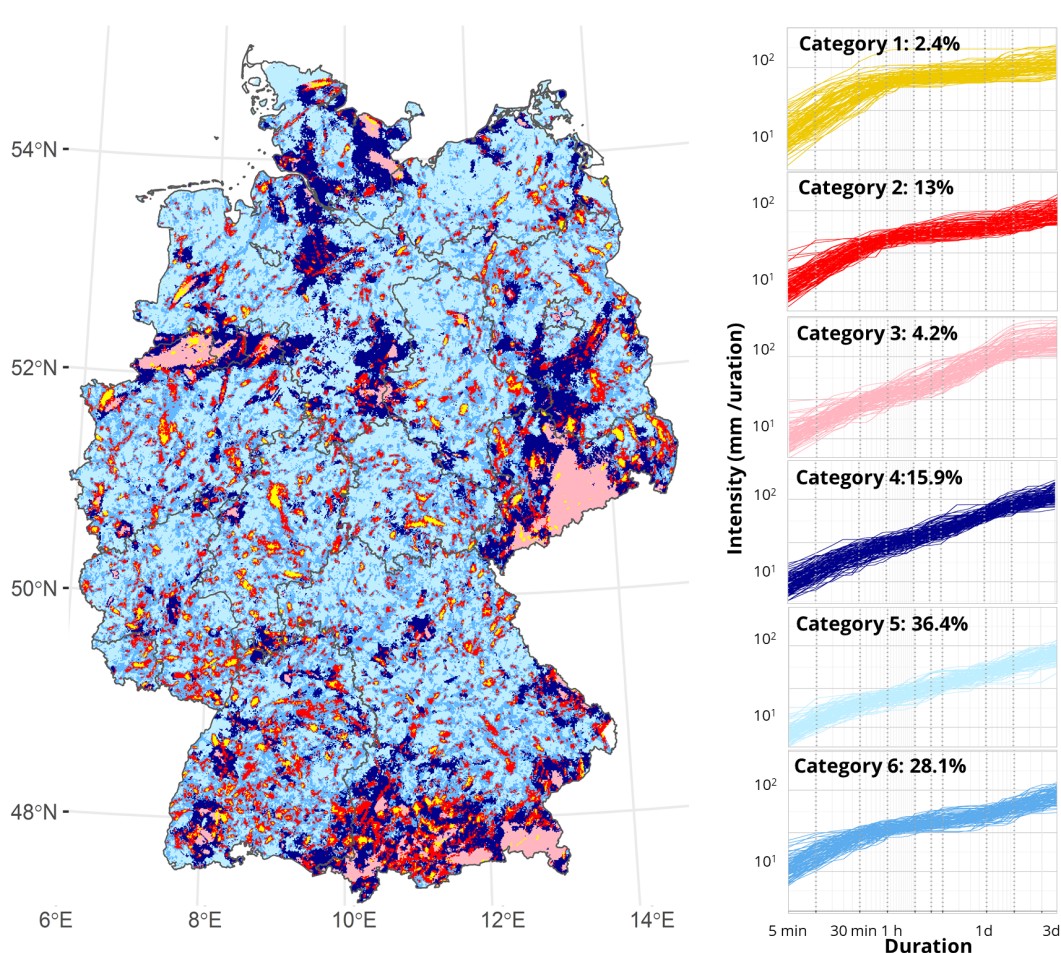

**Figure 9.** Display of the six categories of maximum rainfall depth–duration relationships a) spatially distributed over the whole of Germany, and b) their corresponding curve shapes, sampled at 100 selected radar cells belonging to the corresponding of the six categories.



responsible for the high slope at the beginning part of the curve. Some curves also show another small inflection between 12 h and 1 d that might correspond to an inter storm arrival time over which another large event contributes to the positive slope of the curve at duration from 1 d and longer, or simply contributes to the general high intensity of the whole event. Category 1 cells can be identified as yellow hotspots in Fig. 9 that occur predominantly as smaller cells in the midst of Category 2 (red) and partly in Category 3 (lightpink) cell clusters. Category 2 cells (red) have a similar curve shape as those in Category 1 and

always occur together with Category 1 cells. The curve inflection begins around 30 min and the slope up to 3 d is a little higher than Category 1's slope. This implies that unlike the hotspot locations, those locations experienced strong convective pattern of a little shorter duration, but potentially longer durations in general. Most likely, Category 1 (event center) and Category 2 (event boundary) cells experience local convective events, which are forming in the summer months on warm days with a moist atmosphere. Categories 3 and 4 can be generally associated to large scale events dominated by regional weather patterns

. The three largest clusters in the map can be identified as intense frontal rainfall in August 2002 (Saxony, large cluster in Eastern Germany), heavy downpours over Münster in July 2014 (narrow path in the Northwestern part) and orographic rainfall in the Alpine region of southern Germany. In Category 3, curves show steep slopes up to one day that are abruptly ending with super–daily duration. This category is contributing to the scales between 12 h and 3 d for the curve for whole Germany (Fig. 4). The steep slope at sub–daily duration is because the cells experienced intense convective storms, however, with lower intensity

than Categories 1 and 2. Yet, for daily–scale duration they can experience significant amounts of rainfall. Both Categories 4 and 5, which compose around 50 % of all cells, show rough power law behavior over all scales. Category 4 (dark blue) cells are mainly at the outer borders of the described larger events as well as adjacent cells of Categories 1 and 2. Thus, the curves have steep slopes because the corresponding cells experienced great rainfall. Most cells belong to Category 5 (36 %), showing the smoothest scaling behavior of all categories. These regions/locations never have been hit by any 'extreme' extreme event

that could have altered the power law behavior of the depth–duration relationship. The locations of these cells indicate no spatial pattern and can be seen as a kind of background color of the map. The last Category 6 contains similar characteristics from Categories 1 to 3 and comprises another 30 % of all cells. This category represents cells experiencing common types of convective events with short heavy rainfall sequences on the sub–hourly scale, indicated by a relatively steep slope until 1 h compared to Categories 1 and 2. However, these cells also experience longer rainfall sequences, thus showing an almost "three

phase regime" as the overall curve for Germany with lower values. The found clusters can be further summarized into three classes: Cells that have experienced very heavy rainfall on a sub–hourly scale ( Categories 1 and 2) exhibiting steep slopes at sub–hourly scale and mild slope for longer duration. The second class experienced heavy rainfall sequences of up to 1 d (Categories 3 and 4). The third type shows power law behavior over all scales and can be mainly found in Category 5. Category 6 simultaneously shows characteristics of Classes 1 and 2.

## 3.6  Sensitivity of scaling behavior to ground gauge network density

An important message from Sect. (3.4) and Sect. (3.5) is that the depth–duration relationship at a given point varies location by location based on the rainstorms that the area experienced. This implies that the maximum depth–duration relationship over the entire study area, which is fundamentally the process of the superposition of these various relationships and the picking up



of the very maximum values at each duration, may vary with regard to density and spatial formation of ground gauge networks
(Sect. (3.5)). For this reason, we investigated how the depth–duration relationship would vary with regard to a different number
of sampling pixels. Figure 10 shows the result corresponding to the pixel sample size of 10, 100, 1 000, and 10 000. For
each of the cases, 30 ensembles of random pixel sampling were performed. For each of the plots, the maximum depth–duration
relationship based on all radar pixels was shown for reference. Clear and smooth scaling behaviors are identified when the pixel
sample size is 10 and 100, but the smooth scaling behaviors are lost when including more major rainfall events that formed
the original maximum depth–duration relationship. This emphasizes that the number of rain gauges in a network is extremely
relevant in order to adequately capture rainfall extremes. Note, most scaling relationships of the past including Jennings (1950)
were based on the measurements of the ground gauge network. The station density was used as "best we could get" and was not
tested against a smaller set of stations, as obviously including all reliable extremes improves the relationship. This might work
for Jennings curve as the global scale (sample size over a large space and maximum available time) makes up for the limited
spatial resolution. Regional scaling, however, suffers from the limited spatial extent, which cannot be completely balanced by
a denser network or Radar data.

## 4 Conclusions

A thorough understanding on the scaling behavior of the depth–duration relationship of extreme precipitation has been limited
because its high spatiotemporal variability cannot be fully captured by measurement network composed of limited number
of ground gauges. This study tried to overcome this limitation by using the radar Quantitative Precipitation Estimates (QPE)
rainfall product. The radar QPE enabled clear identification and explanation of the characteristics of the different scaling
regimes of extreme rainfall depth–duration relationships. The maximum depth–duration relationship derived from radar data
did not show clear scaling behavior compared to one based on gauge data from longer time series, but exhibited a "three
phase regime" with a high slope at the duration smaller than 1 h, a plateau at the duration between 1 h and 1 d, and a low
slope at the duration greater than 1 day. The relationship was developed based on only a few extreme rainfall events, which
dominated the shape of the curve and this changed when examining quantiles of pixel maxima. The depth–duration relationship
of lower quantile rainfall (e.g. 99 percentile) showed a smooth scaling behavior and the rainfall events contributing to the curve
sparsely occurred at various locations of Germany. This implies that the modest extreme rainfall events are less sensitive to the
random effects of a limited period (under sampling) and may even share common atmospheric conditions of rainfall generation
regardless of pixel location in a limited region like Germany. The rainfall depth–duration relationship at a single radar pixel did
not show clear power law behavior either. The shape of the curve was governed by the temporal structure of the extreme rainfall
events that the pixel experienced. The point wise clustering of depth–duration relationships revealed three classes of scaling
behavior: a) Linear scaling over all durations, as well as inflections at b) one hour and c) one day, which shows the influence of
small convective cells as well as large scale weather patterns on depth–duration relationship. Therefore, the scaling behavior
at a given pixel location differed significantly location by location because each pixel experienced different extreme rainfall
events. Given that the extreme rainfall depth–duration relationship over a region is a process of overlapping the relationships


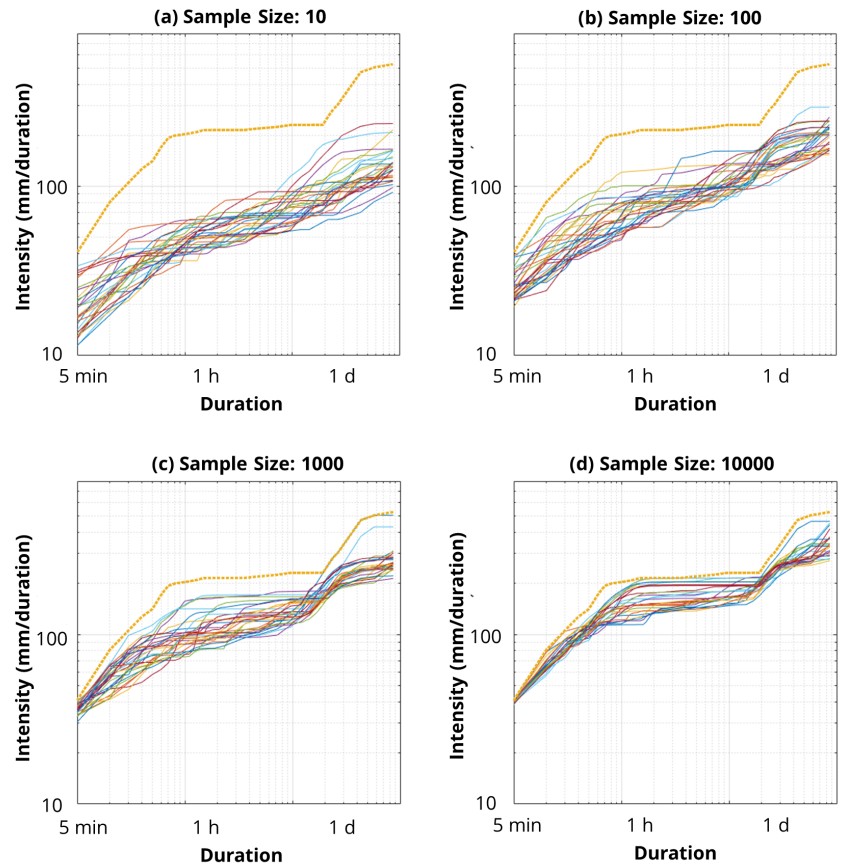

**Figure 10.** Variation of maximum depth–duration relationship with varying sample size of pixels used to develop the relationship

observed at various pixel locations and picking up the highest rainfall values at each duration, the result implies that the depth–duration relationship of extreme rainfall can significantly deviate from power law behavior. With longer available time series of radar in the future, the deviation can be further investigated and tested.

*Code and data availability.*   The data sets used in this study are freely available to download in ASCII and BINARY format and are published under the following DOI: 10.5676/DWD/RADKLIM_YW_V2017.002 (Winterrath et al., 2018b). Analysis was conducted in R (R Core Team, 2019) and Matlab with freely available R-packages ggplot (Wickham, 2016), RasterVis (Perpiñán and Hijmans, 2019), Rcpp-Roll (Ushey, 2018), and fst (Klik, 2019)





*Author contributions.*  JP and DK conceptualized research, developed model code and methodology. JP carried out the analysis together with
DK and RK. DK provided the computing facilities in his lab at Hongik University in South Korea. JP and DK prepared the manuscript with
contribution from all co-authors. CB contributed the scientific idea, contributed to its refinement, to the discussion within the authors' group
and to the writing.

*Competing interests.*  There are no competing interests involved in this study.

*Acknowledgements.*  The authors sincerely acknowledge the financial support by TU Dresden's Institutional Strategy, which is funded by the
Excellence Initiative of the German Federal and State Governments. Professor Kim's contribution was supported by the Korea Environment
Industry & Technology Institute (KEITI) through Water Management Research Program, funded by Korea Ministry of Environment (MOE)
(Project No. 127557). We also thank Dr. Tanja Winterrath from the German Weather Service (DWD) for giving further insight on DWD's
data processing.





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
