# Peer review of "An analysis of temporal scaling behaviour of extreme rainfall in Germany based on radar precipitation QPE data"

_Natural Hazards and Earth System Sciences, 2020_

## Referee Comment (RC1) · Anonymous Referee #1 · 4 Aug 2020

Introduction —————

Extreme rain events may cause severe damage and fatalities and they are of principle interest as they mark the physical limits. This manuscript describes the depth-duration relationship as observed in Germany by the DWD radars (RADOLAN product YW, adjusted by rain gauges) for the period from 2001 to 2016. Although neither the principle method nor the data are new, the application of this method to these data is. The manuscript offers a new sight to Germanys operational radar data and thus has a chance to be published. Nevertheless, it suffers from several drawbacks and misteks making it partly difficult to read and it contains several minor errors that should be

removed before publishing.

Major issues ————

Radar observations cover the complete area of Germany. This advantage is discussed in the paper. Rain gauges may miss the most intense events. Nevertheless, the data quality of radar measurements is spatially variable, depending on the orography and distance from the next radar. Furthermore, radar provides precipitation measurements at a different scale (here 5 min./1 km^2) than rain gauges (commonly 1 min./200 cm^2 in Germany). The paper lacks a discussion on data quality. Especially the shorter extremes might be impacted by ground clutter (in case of 5 min. extremes even from wind turbines or airplanes). With increasing distance the area of each range bin increases, reducing the frequency of extreme values. For a self-contained publication the authors have to describe and discuss these effects. How do they impact the results? Is the spatial distribution of extreme rainfall caused by the precipitation process or by the method of observation?

It should be noted that the radar measurement consist of only one sweep at the lowest undisturbed elevation angle. Scan pattern were variable during the years. (So called precipitation scan.)

Additionally, the authors need to add a short description on the data processing from the measurement to offline quality control and the adjustment with RADOLAN.

Figure 3 shows the "total within-clusters sum of squares". This term is not defined in the paper. Please, describe the procedure to determine the shown curve so it can be comprehended and interpreted.

The authors explain (line 151f) "Between 25 min and 16 h, maximum values are calculated for the southeastern edge of Hesse state in May 29th 2016." This is not reproducible. From Figure 4 we can see, the 25 min extreme is already significantly above 100 mm. In the area between $8.96°$ and $9.37°$ east and $50.15°$ and $50.32°$ north (this

area should cover the location the maximum) rain amunt is below 32 mm. Maximum precipitation on that day is 123.67 mm at 50.54° N, 12.61° E in Ore mountains, Saxony. This maximum is followed by 122.98 mm at 49.22° N, 9.83° E, close to Braunsbach, Baden-Württemberg. That rain event caused estimated 100 million Euro damage and three fatalities, as newspapers reported a few days later. - I did not control the further maxima.

A table is missing, indicating duration, start time, rain amount, and location for each of the blue dots in Figure 4. Without these data no reproduction of the findings is possible.

The value of Figure 6 and its interpretation is not clear for me. What elevates the 3921st greatest event (shown) above the 3920th (not shown)? What is meant by "The lower the quantile, the sparser the location of the quantile rainfall occurence"? What is a sparse location? The location of the dots is totally random, as it is random if an event is the 3920th or 3921st.

I do not get the message, the authors want to transport here. If the focus is on the spatial distribution of extreme events, then show it for some durations (the locations of the strongest 10 (red), 100 (yellow), and 1000 (green) events for a duration of x minutes). If the focus is on the low impact of an individual event, then show the frequency distribution of rain amounts for a certain duration, focusing on the most intense 10000 events or so.

Figure 7 is hard to see, especially in a printed version. Figure 7 is in contradiction to section 3.1. The absolute maximum for 6 hours (Fig. 7d) occurs in the area between Ilmenau and Erfurt, roughly. The 1 hour maximum (Fig. 7c) is in the area of southeast Hesse.

Figure 8: I do not see that Wiesbaden is less fitting to a power law than Stuttgart. The authors do not provide a quantitative or at least objective way to describe the deviation from the power law.

I'm not convinced by the result of the clustering algorithm as shown in Figure 9. I cannot identify clearly distinct properties between the 6 categories. The authors have the same problem and propose to combine the categories into three new categories. The verbal description of the categories remains vague. Probably it was more helpful to state the properties and group the relations/pixels according to predefined criteria. (Cat 1: All relations with more then 40 mm @ 1 hour. Cat 2: less than 40 mm @ 1 hour but more than 100 mm @ 1 day ...) Without a clear description of the categories, Figure 9 lacks a message.

Minor –––

Please be more precise in your wording. A "sample" can be a subset of radar pixels, i.e. only locally restricted. It might also be a temporal subset. A "cell" and a "pixel" refer, as far as I got it, to the same thing: An area of 1 km^2 for which the RADOLAN product provides one rain intensity every 5 minutes. If this is right, please omit one of the two terms. Otherwise define a "cell". You are not talking of storm cells.

There are several issues with the figures: - The y-axis is never precipitation intensity but precipitation sum or precip depth and the unit is mm, not mm/Duration. - Figure 1 does not show the fit for the Spanish measurements and not the individual measurements for the Eastern German measurements. The caption denotes "regional extremes for Germany", the legend "Eastern Germany". Shall this be the same? - Scaling of the x-axes is difficult. In Figure 1 the structure seems to be clear (minutes, hours, days...) but minor tags are missing (they occur as small gaps in the horizontal grid). In Figure 4 the minor tags are too bright to be seen on a printout. In Figure 5 some minor tags have vanished. Could this be unified in a clear visible way?

Line 60f: Lower values of maximum rainfall values on a coarse grid of 400 km^2 gird cells is no underestimation but a known impact of averaging, as Brena-naranjo et al. already mentioned. Whereas "underestimation" indicates a deficit of the measurement or procedure the reduction is physically reasonable.

Line 104: How do you calculate the "imputation bridge"? Radar data are missing so how do you get a rain intensity for these periods?

The description of the methodology is inconsistent and unnecessarily hard to read. E.g. tau is a duration (see line 115), so it might be given in minutes. np is a number, counting the observations at each location. np thus is unitless. What is np-tau+1 (line 124)? What is min? What is h? (line 117). How did you get results for 3-day-extremes when your analyses is limited to 3 hours (line 116)? Line 130 and following do not indicate how you determine B and b. Equation 7 does not fit to equation 6 (somehow tau is lost). Equation 6 might be meant as $\log(M)=B+b*\log(\tau)$, as the figures show log-log axes. Equation 7 then is $M=10^B*\tau^b$ in compliance with Eqn. 1. Besides the mathematical errors there should be more text. E.g.: $M^{(\tau)}_{max,cell}$ (Eqn. 4) are the individual duration-depth relationships for each pixel. This needs to be mentioned.

Figure 4: The publication of WMO, 1994 indicates the value for 3 days should be at 3130 mm (not roughly 4000 mm) and the points at 30 min./200 mm and 3 h/>700 mm are not given there (Table II.5.6). The study of DWD, 2002, cites "Ertel and Schmidt, 1999", as source for their records (without giving a findable reference). DWD 2016 is not accessible. The caption cites a Spanish study but the figure does not show the data.

Figure 7: There are inconsistent color scales, showing 150 mm twice in (c) and (d). The unit should always be mm.

Figure 8, section 3.4: How did you choose the pixels for the cities? All of them are larger than 1 km^2. Please consider to draw all 15 lines of Figure 8 into one pair of axes (different colors, different line style), making it possible to compare the curves.

Line 193: Did you only cluster the 15 depth-duration relations from Figure 8 or from all pixels?

Technical ———

Line 3 and 65: 1 km^2 (it's an area, not a distance)

Line 7: A smooth scaling behavior was (not were).

Line 20/22: If probable maximum precipitation is abbreviated by PMP you should introduce this abbreviation: Probable maximum precipitation (PMP)...

Line 35ff: Bad deal with units! (1) is a numerical value equation, thus you need to indicate units of all variables.

Line 37: In (1) duration is d not D.

Line 37: Are the values 425 and 0.47 found by Jennings (1950)? If so how did that curve change during the last 70 years? Shouldn't it be "alpha and beta had the values", as climate change might have changed these values?

Line 77: DWD is running (not has been running) a radar network. They still do.

Line 85: There is no need to mention RW product. There are more RADOLAN products but RW and YW.

Line 114: What is a radar cell? If a pixel is meant, call it a pixel. Are you aware of the difference of a pixel (beam volume element) within the basis data (sphere coordinates, growing size with distance) and a RADOLAN grid element?

Line 135: Please, add "Eqn. (4)" to the first sentence.

Line 143: There is no blue solid line in Figure 4. There is no red dotted line in Figure 4.

Line 146: As well as for one (not some) sub-daily values. Only for 1 h radar has a higher value than rain gauges.

Figure 5: Please consider to add the curves of this figure into Figure 4 and to remove the inlay with the map of Germany into a new figure.

Line 169: Which curve? There is no reference.

Line 185: Stuttgart, not Stuttugart

Line 193: depth-duration (not dept-duration)

---

## Author Comment (AC1) · 8 Sep 2020

**nhess-2020-192: An analysis on temporal scaling behavior of extreme rainfall of Germany based on radar precipitation QPE data (Pöschmann et al.)**

**Reply to the comments from Referee #1:**

We are thankful for referee #1's constructive and very detailed comments and suggestions that helped us to improve the original draft. We have provided responses to all comments (in blue) and updated the manuscript according to the suggestions. All line numbers given in our responses refer to the original version of the manuscript in order to avoid confusion.

**Comments:**

**Introduction**

Extreme rain events may cause severe damage and fatalities and they are of principle interest as they mark the physical limits. This manuscript describes the depth-duration relationship as observed in Germany by the DWD radars (RADOLAN product YW, adjusted by rain gauges) for the period from 2001 to 2016. Although neither the principle method nor the data are new, the application of this method to these data is.

The manuscript offers a new sight to Germanys operational radar data and thus has a chance to be published. Nevertheless, it suffers from several drawbacks and misteks making it partly difficult to read and it contains several minor errors that should be removed before publishing.

**Major issues**

1. Radar observations cover the complete area of Germany. This advantage is discussed in the paper. Rain gauges may miss the most intense events. Nevertheless, the data quality of radar measurements is spatially variable, depending on the orography and distance from the next radar. Furthermore, radar provides precipitation measurements at a different scale (here 5 min./1 km^2) than rain gauges (commonly 1 min./200 cm^2 in Germany). The paper lacks a discussion on data quality. Especially the shorter extremes might be impacted by ground clutter (in case of 5 min. extremes even from wind turbines or airplanes). With increasing distance the area of each range bin increases, reducing the frequency of extreme values. For a self-contained publication the authors have to describe and discuss these effects. How do they impact the results? Is the spatial distribution of extreme rainfall caused by the precipitation process or by the method of observation? It should be noted that the radar measurement consist of only one sweep at the lowest undisturbed elevation angle. Scan pattern were variable during the years. (So called precipitation scan.) Additionally, the authors need to add a short description on the data processing from the measurement to offline quality control and the adjustment with RADOLAN.

**Response:** The data quality has indeed been a big concern during the data processing and evaluation of the results. Our biggest concern had been a potentially high number of outliers due to radar errors as have been observed with the RADOLAN products, which would result in too high values for rainfall maxima. In this study, the focus is on the post-processed data of RADKLIM. The RADKLIM data should be significantly and consistently improved compared to RADOLAN. Therefore, we included more details on the data quality of RADKLIM.

Lines 81 – 84 were edited as follows:

"Since the quality enhancement of RADOLAN is ongoing without post-correcting previous data, the so–called radar climatology project of the DWD, RADolanKLIMatologie (RADKLIM,Winterrath et al., 2017) has consistently reanalyzed the complete radar data archive set since 2001 to attain homogeneity of the data that were processed through different algorithms. Compared to RADOLAN, RADKLIM has implemented additional correction algorithms that lead to much more plausible spatial distribution of precipitation totals including fewer typical radar artefacts, improved representation of orography as well as efficient correction of range-dependent path-integrated attenuation at longer time scales (Kreklow et al., 2019). Whereas RADOLAN is not suited for climatological applications and aggregated precipitation statistics, RADKLIM is a promising data set for these kinds of applications. The RADKLIM data is available .."

Lines 91 – 94 were edited as follows:

"The YW product covers the area composed of 1100 x 900 pixels with the spatial resolution of 1 km (improved compared to former version of RADOLAN). Remaining weaknesses of RADKLIM (as outlined in Kreklow et al. (2019)) are a higher number of missing values (compare below) than for RADOLAN as well as an overall negative bias causing a rather "underestimation" of high intensity rainfall due to spatial averaging and rainfall-induced attenuation of the radar beam."

In addition, we combined sections 2.1.1 and 2.1.2 into 2.1. (by removing their subtitles). Furthermore, we added text in section 3.1., explaining the possible underestimation of sub-hourly values.

The following sentence was added at line 151 (center):

"As mentioned in the data quality description, it is possible that these sub-hourly values do not represent the "real" German-wide extremes for 2001-2016 since very short durations are specially effected by averaging effects of the radar processing."

The uncertainty of radar-based products remains a challenge for the coming years and the product quality and characteristics obviously have an effect on the distribution and intensity of extreme values of rainfall. However, a detailed discussion of the validity of the data set is not the scope of this study, rather to discuss how this dataset reflects well-documented features of rainfall extremes from the "pre-radar world". RADKLIM is currently the best consistent representation of rainfall for the whole of Germany (according to DWD). Thus, we purposely used the dataset despite unknown (as always) deficiencies. As the results appear to be plausible and consistent, the discussion deals with the RADKLIM extremes as observed extremes. Generally said, almost half a million pixels will provide a good representation of the "true" characteristics.

Following sentence was added at the end of the conclusion:

"Also, the known "underestimation" of rainfall extremes by RADKLIM-YW and the potential impact on the results needs further evaluation."

2. Figure 3 shows the "total within-clusters sum of squares". This term is not defined in the paper. Please, describe the procedure to determine the shown curve so it can be comprehended and interpreted.

**Response:** We changed the passage in order to make it more comprehensible.

Edited text passage (from line 138):

"If the number of clusters is not predefined, it can be identified by drawing an elbow chart as seen in Fig. 3. For different number of clusters K the measure of the variability of the observations within each cluster (Total within-cluster sum of squares, y-axis) is calculated and the curve should bend like an elbow at the optimal value. Since the algorithm did not suggest a number of clusters, we chose six clusters for a sufficiently detailed analysis since it gave consistent results when repeating the automatic algorithm for several time (each time the algorithm clusters slightly different)."

3. The authors explain (line 151f) "Between 25 min and 16 h, maximum values are calculated for the southeastern edge of Hesse state in May 29th 2016." This is not reproducible. From Figure 4 we can see, the 25 min extreme is already significantly above 100 mm. In the area between 8.96° and 9.37° east and 50.15° and 50.32° north (this area should cover the location the maximum) rain amount is below 32 mm. Maximum precipitation on that day is 123.67 mm at 50.54° N, 12.61° E in Ore mountains, Saxony. This maximum is followed by 122.98 mm at 49.22° N, 9.83° E, close to Braunsbach, Baden-Württemberg. That rain event caused estimated 100 million Euro damage and three fatalities, as newspapers reported a few days later. - I did not control the further maxima. A table is missing, indicating duration, start time, rain amount, and location for each of the blue dots in Figure 4. Without these data no reproduction of the findings is possible.

**Response:** We want to thank the referee for carefully checking our results. At the same time, we apologize for our mistake. In Figure 4, locations and rain amounts are a correct representation of what we get from the data, but the identification of dates went very wrong. We changed that, updated the text and added a table indicating the suggested characteristics. From the table the findings are now reproducible. The Binary RADKLIM-YW files are available at the DWD online depository (https://opendata.dwd.de/climate_environment/CDC/grids_germany/5_minutes/radolan/reproc/2017_002/bin/) for the given dates and the locations are identifiable by their WGS84 coordinates.

The text is edited as follows (from line 151):

"Between 25 min and 16 h, maximum values are calculated for a location at the border of Hesse state and Bavaria in August 25th 2006, which has not been documented in public news. The extreme event around September 29th/30th 2003 around Berlin comprised the maximum depth-duration relationship at the duration between 18 h and 2 d."

The following Table 1 is inserted into the manuscript:

**Table 1.** Rainfall records for different duration from RADKLIM-YW for 2001 - 2016 with corresponding locations.

| Duration | Start Date | Start Time (Time Zone: Berlin) | Precipitation Sum (mm) | Location (WG84) |
|---|---|---|---|---|
| 5 min | 2009-07-04 | 2:10 PM | 40.94 | 48.50015° N, 9.35161° E |
| 10 min | 2006-07-07 | 9:30 AM | 80.82 | 51.22436° N, 8.767699° E |
| 15 min | 2010-07-12 | 11:05 PM | 105.61 | 52.79713° N, 12.39296° E |
| 20 min | 2002-07-30 | 5:15 PM | 127.32 | 48.82225° N, 9.577044° E |
| 25 min-16 h | 2006-08-25 | 05:25 AM - 1:25 PM | 141.13-230.67 | 50.21148° N, 9.201292° E |
| 18 h-1 d | 2003-09-29 | 09:05 AM-03:05 PM | 258.91-327.45 | 52.52761° N, 13.5271° E |
| 1.5-2 d | 2003-09-28 | 02:20-9:35 PM | 471.67-503.66 | 52.52761° N, 13.5271° E |
| 3d | 2001-04-08 | 06:50 AM | 525.89 | 53.67822° N, 10.00056° E |

Maxima of 25min - 16h as well as from 18h - 2d correspond to the same location and date and are thus summarized.

4. The value of Figure 6 and its interpretation is not clear for me. What elevates the 3921st greatest event (shown) above the 3920th (not shown)? What is meant by "The lower the quantile, the sparser the location of the quantile rainfall occurrence"? What is a sparse location? The location of the dots is totally random, as it is random if an event is the 3920th or 3921st. I do not get the message, the authors want to transport here.

**Response:** We are sorry that the interpretation of the Figure is unnecessarily difficult. The purpose of placing the Figure like this is related to Figure 5: We want to show that if not taking the maxima of maxima, but certain quantiles of maxima (we chose to take numbers that correspond to 99.999%, 99.99 %, 99.9% and 99%, but obviously other values could have been chosen):

1) No longer seven locations (as in Figure 4) hold all maxima, but that the number of locations increase and also spread over all of Germany.

2) Despite 1), the corresponding depth-duration relationships are straightening out (=getting smoother) and start to reflect rather natural conditions of rainfall (e.g. dominance of Alpine region in Figure 6d) instead of singular "extremes of extremes".

We simplified the sentences around the irritating sentence (l. 164) that the referee pointed out in order to make it clearer.

Text edition (from l. 164):

"It shows that the number of locations increases the lower the quantile of maximum rainfall is. This suggests the reduction of the influence of one single rainfall event on the depth-duration relationship causing inflection in the curve. Additionally, from a certain degree of quantile (Fig. 6 d) the locations of maximum rainfall contributing to the development of the rainfall-duration relationship seem to happen mainly in the wider Alpine region in South Germany. This suggests that rather natural rainfall mechanisms are dominating the scaling relationship, such as regional characteristics and meteorological conditions (e.g. orographic lifting or leewards effects). Naturally, one would assume that this heterogeneity of the meteorological conditions and rainfall generating mechanisms will reflect rather regional characteristics and will exhibit some irregular scaling behavior. Contrary to this conjecture, the curves in Fig. 5 (99.9% and 99%) show a rather smooth scaling behavior" [end of subsection 3.2]

5. If the focus is on the spatial distribution of extreme events, then show it for some durations (the locations of the strongest 10 (red), 100 (yellow), and 1000 (green) events for a duration of x minutes). If the focus is on the low impact of an individual event, then show the frequency distribution of rain amounts for a certain duration, focusing on the most intense 10000 events or so.

**Response:** We are grateful for the suggestions by the referee, however chose not to change the Figure since it would change our message (spatial distribution of extremes is already shown in Figure 7).

6. Figure 7 is hard to see, especially in a printed version. Figure 7 is in contradiction to section 3.1. The absolute maximum for 6 hours (Fig. 7d) occurs in the area between Ilmenau and Erfurt, roughly. The 1 hour maximum (Fig. 7c) is in the area of southeast Hesse.

**Response:** We understand the concern of the respected referee. The purpose of Figure 7 is not to identify the location of maxima of maxima, since they are already provided in Figure 4. We

want to show that the longer the duration, the more "organized" the maxima get and big events with long and heavy rainfall are visible on the map as well as seasonal and terrain-related patterns would most-likely become visible. Figure 7 focuses on the overall distribution of maxima. Singular pixels of maxima would be too hard to trace on a small map without zooming in, as the referee pointed out. If the referee would find it more suitable we could change the color scheme as given in the following example (colors represent quantiles):

[Figure]

7. Figure 8: I do not see that Wiesbaden is less fitting to a power law than Stuttgart. The authors do not provide a quantitative or at least objective way to describe the deviation from the power law.

**Response:** We thank the referee for pointing out this important issue. We placed a reference line in all Figures in order to make the differences more visible for the readers. It is true, that the deviation is a subjective interpretation from the authors of the study. We changed the text from l. 185 a little bit after taking again a look at the Figure with reference line, since we agree that Wiesbaden is not different to the mentioned cities Hannover, Kiel, etc.

Text edition from line 184:

"Figure 8 shows the maximum rainfall–duration relationship of the radar pixels at the major cities of Germany with a blue line as reference to see the differences better. Except for Hamburg and Stuttgart, most cities exhibit slight (Hannover, Kiel, Magdeburg, Potsdam, Schwerin, Wiesbaden) to considerable ....."

New Figure 8:

[Figure]

**Figure 8.** Maximum depth–duration relationships at rain gauge locations of major cities of Germany

8. I'm not convinced by the result of the clustering algorithm as shown in Figure 9. I cannot identify clearly distinct properties between the 6 categories. The authors have the same problem and propose to combine the categories into three new categories. The verbal description of the categories remains vague. Probably it was more helpful to state the properties and group the relations/pixels according to predefined criteria. (Cat 1: All relations with more then 40 mm @ 1 hour. Cat 2: less than 40 mm @ 1 hour but more than 100 mm @ 1 day …) Without a clear description of the categories, Figure 9 lacks a message.

**Response:** The purpose of the clustering was to automatically group pixels of similar depth-duration relationship characteristics. Thus, we cannot predefine properties before, but evaluate the clustered groups. As the referee points out, we propose to combine the 6 clusters into 3 categories, since we find it more meaningful.

The authors think that Figure 9 proves that the maximum depth-duration relationship for all pixels is mainly driven by singular events, whereas the lower the maxima we look at, the more smooth the curve will get. We put the Category 5 regression line into the sub-figures of all other Categories of Figure 9 in order to make the difference more visible.

New Figure 9:

[Figure]

**Figure 9.** Display of the six categories of maximum rainfall depth–duration relationships a) spatially distributed over the whole of Germany, and b) their corresponding curve shapes, at 100 randomly selected radar pixels belonging to the corresponding of the six categories.

The following was edited (Line 195):

"Figure 9 shows a categorical map of Germany representing each category with a certain color. Additionally, depth-duration relationships at 100 randomly chosen grid elements from each category are shown with the regression line from Category 5 as reference."

**Minor**

9. Please be more precise in your wording. A "sample" can be a subset of radar pixels, i.e. only locally restricted. It might also be a temporal subset.

**Response:** The authors generally think that the meaning of "sample" as subset of radar pixels or temporal subset is clear in the corresponding context. However, we have revised all occurrence of sample and changed it at few places.

10. A "cell" and a "pixel" refer, as far as I got it, to the same thing: An area of 1 km^2 for which the RADOLAN product provides one rain intensity every 5 minutes. If this is right,

please omit one of the two terms. Otherwise define a "cell". You are not talking of storm cells.

**Response:** We agree that we should use consistent wording and chose to use "pixel". We changed it in the whole manuscript.

11. **There are several issues with the figures**: - The y-axis is never precipitation intensity but precipitation sum or precip depth and the unit is mm, not mm/Duration. -

**Response:** There exist representations of the depth-duration relationships where the precipitation sum is provided as intensity. However, we agree that providing the precipitation sum with unit mm is more meaningful and we changed it in Figures 1, 4, 5, 8, 10.

12. Figure 1 does not show the fit for the Spanish measurements and not the individual measurements for the Eastern German measurements. The caption denotes "regional extremes for Germany", the legend "Eastern Germany". Shall this be the same?

**Response:** We changed "Eastern Germany" to "Reg. Extremes Germany" to be more consistent. For consistency reasons we removed the individual values and only showed the fit of the measurements, since no individual values are available for the German regional curve.

13. Scaling of the x-axes is difficult. In Figure 1 the structure seems to be clear (minutes, hours, days...) but minor tags are missing (they occur as small gaps in the horizontal grid). In Figure 4 the minor tags are too bright to be seen on a printout. In Figure 5 some minor tags have vanished. Could this be unified in a clear visible way?

**Response:** We added minor tags to Figure 1 and added "minutes", "hours" and "days" to Figures 4 and 5 and made the minor tags for Figures 4 and 5 more visible.

14. Line 60f: Lower values of maximum rainfall values on a coarse grid of 400 km^2 gird cells is no underestimation but a known impact of averaging, as Brena-naranjo et al. already mentioned. Whereas "underestimation" indicates a deficit of the measurement or procedure the reduction is physically reasonable.

**Response:** The authors think that the word underestimation can also be a logical consequence of averaging and the explanation is given in the following sentence in our opinion. However, we respect the referee's concern and changed the two sentences (from line 60f):

"They showed that the maximum of the areal rainfall averaged over the ~20 km x ~20 km data grid has the scaling exponent of ~0.43 which is similar to that of Jennings (1950). However, the coarse spatial resolution of the satellite data easily misses the small scale rainfall variability that is closely associated with extreme values, thus the found extremes in the satellite data are lower than expected (Cristiano et al., 2017; Fabry, 1996; Gires et al.,2014; Kim et al., 2019; Peleg et al., 2013, 2018)."

15. Line 104: How do you calculate the "imputation bridge"? Radar data are missing so how do you get a rain intensity for these periods?

**Response:** As it is written in the text, the imputation bridge is the "maximum rainfall difference between right BEFORE and AFTER a data gap". We thus do not need rain intensity for these periods, but calculate how much would need to be imputed for the highest differences in intensities. The argument would be, that when values before and after a data gap are quite "similar", most likely no big changes happen in between and if there is a big difference (=high imputation bridge), it is higher likely that we do not know what happens between the two values.

However, as we have further pointed out, the imputation of NA values for rainfall events imposes too much uncertainties regarding the analysis of extremes.

16. The description of the methodology is inconsistent and unnecessarily hard to read. E.g. tau is a duration (see line 115), so it might be given in minutes. np is a number, counting the observations at each location. np thus is unitless. What is np-tau+1 (line 124)? What is min? What is h? (line 117). How did you get results for 3-day-extremes when your analyses is limited to 3 hours (line 116)? Line 130 and following do not indicate how you determine B and b. Equation 7 does not fit to equation 6 (somehow tau is lost). Equation 6 might be meant as log(M)=B+b*log(tau), as the figures show log-log axes. Equation 7 then is M=10ˆB*tauˆb in compliance with Eqn. 1. Besides the mathematical errors there should be more text. E.g.: Mˆ(tau)_max,cell (Eqn. 4) are the individual duration-depth relationships for each pixel. This needs to be mentioned.

**Response:** The authors agree that the methodology part might be unnecessarily detailed creating more confusion potentially. Thus, we shortened sections 2.2.1 and 2.2.2 into one paragraph 2.2. "Depth-Duration relationships" and removed most of the equations for a better reading.

The methodology section is now shortened as follows:

"2.2 Depth-Duration relationships

Maximum rainfall values for each duration τ between 2001-2016 were calculated with rolling sums applied over moving windows using the R package RcppRoll (Ushey, 2018). Durations of up to 3 d were chosen for the analysis, with multiple steps for minutes and hours out of our interest for sub–hourly and sub–daily pattern. The records may include non-rainfall data thus do not imply continuous precipitation for the period considered. Values were not aggregated spatially, since this usually reduces the maximum intensity values (Cristiano et al., 2018).
First, the extreme values for each pixel and duration $M_{max}^{\tau,pixel}$ are calculated. Afterwards, the overall maxima for whole Germany for each τ ($M_{max}^{(\tau)}$) is extracted from these calculated extreme values. Based on these results, the depth–duration relationships can be build for each pixel as well as for the whole of Germany."

Section 2.2.3 is thus changed to 2.3.

min and h (line 117) are the journal's abbreviations for minute and hour, but we have written it out as "minutes and hours" now.

"3 h" in line 116 was mistakenly written and was replaced with "3 d" (for days).

17. Figure 4: The publication of WMO, 1994 indicates the value for 3 days should be at 3130 mm (not roughly 4000 mm) and the points at 30 min./200 mm and 3 h/>700 mm are not given there (Table II.5.6). The study of DWD, 2002, cites "Ertel and Schmidt, 1999", as source for their records (without giving a findable reference). DWD 2016 is not accessible. The caption cites a Spanish study but the figure does not show the data.

**Response:** The authors added NWS 2016 (https://www.nws.noaa.gov/oh/hdsc/record_precip/record_precip_world.html) as a reference for the world records which provides updated values. The 3 days value of 3929 mm corresponds to an event in La Réunion in 2007. The values for 30 mins and 3 h (724 mm, USA 1942) are shown in the graph, which might be better visible after we have changed the x-axis. However, we

mistakenly twisted numbers for the 30 min value and show 208 mm instead of the correct value of 280mm. We updated Figure 4 (see attachments Fig. 3)

[Figure]

**Figure 4.** Maximum depth–duration relationships and locations of rainfall maxima. Dots: German radar derived data of this study. Non-filled triangles: German ground network (Rudolf and Rapp, 2003; DWA, 2015; DWD, 2020). Filled Triangles: World records (World Meteorological Organization, 1994; NWS, 2014). Map: Locations of rainfall maxima (based on the German radar data) for the considered duration.

We contacted the DWD about accessing the DWD 2016 source and they informed us, that the same Table from DWD 2016 is also available in "DWD (2020): Nationaler Klimareport. 4. korrigierte Auflage, Deutscher Wetterdienst, Potsdam, Deutschland, 54 Seiten. DWD. Stand Druckversion: 04/20 " (p. 37) accessible via https://www.dwd.de/DE/leistungen/nationalerklimareport/download_report_auflage-4.pdf?__blob=publicationFile&v=11. We changed the reference accordingly.

We agree to replace the study DWD 2002 with the report of B. Rudolf and J. Rapp (2003), "Das Jahrhunderthochwasser der Elbe: Synoptische Wetterentwicklung und klimatologische Aspekte" (values provided in Table 2), accessible via

https://pdfs.semanticscholar.org/fdfc/a0eb2c7ac37d2d80ddd2700b3f710a7fed79.pdf, with no cross-reference of another study, but the source is the DWD.

The Spanish study indeed is not included in the Figure, so we removed it in the caption.

18. Figure 7: There are inconsistent color scales, showing 150 mm twice in (c) and (d).

**Response:** When comparing to Figure 4, it is clear that the maximum values do not change significantly between 1 hour and 6 hours. Thus, the range of values remains similar for both Figure 7 (c) and (d). However, when looking at Figure 7 (d) we can observe that more pixels have higher values than for Figure 7 (c), which is what we would expect, since more rainfall is happening for longer time periods.

19. The unit should always be mm.

**Response:** Though we think that it would be clearer to use mm/duration of the Figure, we changed all units to "mm".

20. Figure 8, section 3.4: How did you choose the pixels for the cities? All of them are larger than 1 km^2. Please consider to draw all 15 lines of Figure 8 into one pair of axes (different colors, different line style), making it possible to compare the curves.

**Response:** We added a reference line in Figure 8 in order to compare them with each other (compare updated Figure in comment 7.). Drawing all 15 lines into one plot will result into a rather chaotic display in our opinion.

Rain gauge data can be downloaded from a DWD repository including a list of all rain gauge stations with x-y coordinates. We took these coordinates and transformed them into the RADKLIM/RADOLAN projection. Since we are reprojecting point data (gauge) and match it with raster data (RADKLIM) there might be very slight geographic shifts between gauge locations and the corresponding 1 km² raster grid pixel. We included a half-sentence in Figure 8 (compare comment 7).

21. Line 193: Did you only cluster the 15 depth-duration relations from Figure 8 or from all pixels?

**Response:** We added a half-sentence that Figure 8 clustering is for all pixels as described in the corresponding section 2.2.3

Edition of first sentence in line 193:

"The maximum depth–duration relationships for all pixels within Germany were clustered since Fig. 8 indicated that they might show similar shapes."

**Technical**

22. Line 3 and 65: 1 km^2 (it's an area, not a distance)

**Response:** We changed it.

23. Line 7: A smooth scaling behavior was (not were).

**Response:** We changed it.

24. Line 20/22: If probable maximum precipitation is abbreviated by PMP you should introduce this abbreviation: Probable maximum precipitation (PMP)...

**Response:** We changed it according to the suggestion.

25. Line 35ff: Bad deal with units! (1) is a numerical value equation, thus you need to indicate units of all variables.

**Response:** We changed it.

26. Line 37: In (1) duration is d not D.

**Response:** We changed it to D in all of the occurrences.

27. Line 37: Are the values 425 and 0.47 found by Jennings (1950)? If so how did that curve change during the last 70 years? Shouldn't it be "alpha and beta had the values", as climate change might have changed these values?

**Response:** The values found by Jennings remained surprisingly stable over the last 70 years, however, Gonzales et al. (2017) updated the envelope for the world records and we included the update in the paragraph.

Text edition for comments 25. – 27. (from line 33):

"Jennings discovered that this unique scaling behavior holds at the rainfall duration between 1 min through 24 months. Paulhus (1965) showed that the same power law relationship holds after addition of new world rainfall record observed at the island of La Réunion at the duration between 9 h and 8 d. The envelope for this extreme values can be expressed as:

$$P = \alpha D^{\beta} \quad (1)$$

where P is the maximum precipitation (in mm) occurring in duration D (in h), the coefficient α (425 in Paulhus (1965)) represents the value at one hour of the depth–duration relationship plotted on the log–log plane, and the exponent β (0.47 in Paulhus (1965)) is the parameter characterizing the scaling behavior of the depth–duration relationship. The Spanish study of Gonzales and Bech (2017) updated the world envelope's slope to 0.51, showing a remarkable stability. Multiple exponents describing the scaling property of ......."

28. Line 77: DWD is running (not has been running) a radar network. They still do.

**Response:** We changed it.

29. Line 85: There is no need to mention RW product. There are more RADOLAN products but RW and YW.

**Response:** Yes, there are several RADOLAN products available from the DWD. However, RADKLIM-RW and YW are products from a separate project from the DWD as mentioned in the section. We feel the need to mention the RADKLIM-RW product, since the product that we used (RADKLIM-YW) is more or less directly derived from this hourly product and not an independently calibrated/adjusted product.

30. Line 114: What is a radar cell? If a pixel is meant, call it a pixel. Are you aware of the difference of a pixel (beam volume element) within the basis data (sphere coordinates, growing size with distance) and a RADOLAN grid element?

**Response:** In this study we are talking about a RADKLIM grid element as defined by the referee before: "An area of 1 km^2 for which the RADOLAN (RADKLIM) product provides one rain intensity every 5 minutes." We follow the referee's suggestion as written as reply to comment 10 and consistently used the word "pixel" to avoid further confusion.

31. Line 135: Please, add "Eqn. (4)" to the first sentence.

**Response:** We added it.

32. Line 143: There is no blue solid line in Figure 4. There is no red dotted line in Figure 4.

**Response:** We changed it to the characteristics shown in Figure 4: Triangles (filled/empty) and dots

33. Line 146: As well as for one (not some) sub-daily values. Only for 1 h radar has a higher value than rain gauges.

**Response:** We changed the wording according to the referee's suggestion.

34. Figure 5: Please consider to add the curves of this figure into Figure 4 and to remove the inlay with the map of Germany into a new figure.

**Response:** We are thankful for the suggestion, however, prefer to keep the Figures separate since Figure 4 focuses on the maxima of maxima, whereas Figure 5 is more related to Figure 6 and does transport a different message than Figure 4.

35. Line 169: Which curve? There is no reference.

**Response:** We added the reference. It is the two bottom lines of Figure 5. (compare response to comment 4, final sentence in the text edition")

36. Line 185: Stuttgart, not Stuttugart

**Response:** We changed it.

37. Line 193: depth-duration (not dept-duration)

**Response:** We changed it.

---

## Referee Comment (RC2) · Anonymous Referee #2 · 27 Oct 2020

**Review of **An analysis on temporal scaling behavior of extreme rainfall of Germany based on radar precipitation QPE data** by Pöschmann et al.**

October 27, 2020

**General Comments**

The authors present an analysis of precipitation intensity-duration relationships over Germany based on the RADKLIM radar dataset, with a spatial resolution of 1 km$^2$ and a temporal resolution of 5 minutes. They find a non-smooth scaling relationship, with indications of regime transitions between different temporal aggregation lengths. I think the study is interesting and I'm not aware of a similar study using radar data over Germany. I didn't see any fundamental flaws in the work and therefore think it could be publishable.

Before it can be published though, I think there is room for improvement. This mostly relates to the text and presentation, rather than the science. In some parts the manuscript can be a bit confusing and hard to follow, with aspects explained in a sub-optimal manner. I've outlined my suggestions below.

I should note that, in order to perform an independent and unbiased review, I refrained from reading the already published reviewer comment and therefore apologise for any repetition of what may have already been said!

**Main Comments**

**1. The "three-regime" scaling curve.** It wasn't clear to me if the "three-regime" scaling curve you report is a new finding or not, i.e. is there any other literature which report a multiple-regime scaling curve? If the three-regime scaling curve is a novel result, then you should emphasize this. If it is not, then you should cite other studies where multiple-regime scaling curves were reported.

**2. The RADKLIM data set.**

**(a)** As the whole study and its results hinge on the RADKLIM dataset, I think the reader needs to be given more information about this dataset and its limitations, particularly how they might affect the results of an extreme precipitation study. This is particularly important because there is no documentation for the RADKLIM dataset available in English. As far as I know, the only available source is DWD Report No. 251, which is in German (`https://www.dwd.de/DE/leistungen/pbfb_verlag_berichte/pdf_einzelbaende/251_pdf`).

For example, RADAR data are known to often contain artefacts due to interference (wind turbines, WLAN networks, etc.), which can be particularly problematic when looking at intense events. What steps were taken in the production of RADKLIM to eliminate or reduce such artefacts? You'll find all the necessary information in Section 4 of DWD Report No. 251 (see link). Of course we can't expect the data and results to be perfect, so we need to transparently present these issues to the reader to help them form their own opinions.

**(b)** The scaling regime transitions at 1 hour and 1 day got me thinking. As we know, RADKLIM uses station data to adjust radar-measured precipitation. According to DWD Report No. 251 (Sections 4c and 4d), this is done with hourly station data where available. If no hourly station data are available, then daily station data are used. The number of hourly and daily stations used can be seen in Fig. 5 of the aforementioned report. The radar data are summed to the temporal resolution of the station data and adjusted, before a "disaggregation" procedure is applied to return the radar data to their original temporal frequency.

Could it be that the different regimes result from this adjustment process? Maybe those pixels adjusted with hourly gauge data tend towards a scaling curve with one characteristic slope, while those pixels adjusted with daily gauge data tend towards a scaling curve with a different characteristic slope? What do the authors think? I think this question underlines the importance of my first main point about whether your multiple-regime scaling curve is a unique finding or not. If a multiple-regime scaling curve has never been reported before (even in radar-based studies), then it would arouse concern that your three-regime scaling curve may be a data artefact. If multiple-regime scaling curves are common, then my comment can likely be ignored.

**3. Figure Captions.** I think it would help the readers if the figure captions were a bit more descriptive. Generally, they are just one sentence. For example, in the captions you could add some more text highlighting the interesting aspects of the figures, so that it is clear to the reader what exactly the motivation for showing the plot is and why the presented result is interesting. This saves the reader from having to flip back and forth between the text and image (which may be several pages apart).

**Minor Comments and Technical Corrections**

-Language: As far as I know, NHESS publishes using British English. The manuscript currently uses American English spellings. If you change the language of your spell-checker to British English you should easily be able to find all of these misspellings. For example, "behavior → behaviour", "modeled → modelled", "color → colour", "neighboring → neighbouring", etc.

-Superfluous text: I think there's a fair bit of superfluous text which could be eliminated. As just a small example, I don't think anyone is interested that (L78) the DWD "is providing different free and purchasable rainfall data derived from it". Maybe I'm being a bit picky here, so you can ignore my comment if you want! Less redundant text is, in general, always appreciated by the reader.

-Title: "An analysis *of* temporal scaling behavior of extreme rainfall *in* Germany based on radar precipitation QPE data" (not 'on' or 'of')

-L38: Here you've switched from "d" to "D".

-L55-: For your discussion of the impact of sparse rain gauge networks (also elsewhere in the manuscript), the very new publication of Lengfeld et al. (2020) based on the RAD-KLIM network may be particularly interesting for you.

-L88: What are RADOLAN-RY and RADOLAN-RH?

-Eq. 7: Something has gone wrong here. If $\log(M) = B + b \cdot \tau$, then $M = 10^{(B+b \cdot \tau)}$. Also, there's an open bracket in Eq. 6.

-L121: Please state how many pixels exist for the whole of Germany, i.e. N = ? This is also useful to know when we look at the subsamples in Fig. 10. I therefore also suggest repeating the value of N in the caption of Fig. 10 (like you show in Fig. 5) and also somewhere around the line 235-238.

-L138: There's an open bracket here too.

-Fig. 3: Personally I think that Fig. 3 is superfluous. You could just state the result in one sentence without showing the plot, and take up less space. k-Means and elbow plots are pretty standard and are unlikely to confuse readers. Do you know that the final publication charge will be based on the number of pages? Alternatively, if you really like the plot you could put it in a supplementary info file or an appendix.

-L143 and Fig. 4: I can't see any "blue solid lines" or "red dotted lines" in Fig. 4! I only see blue dots, black triangles and empty triangles. Also, shouldn't the unit in Fig. 4 y-axis just be mm? The caption for Fig. 4 is also confusing, because it talks about "Spanish ground gauge records" but these aren't visible in the plot.

-Fig. 6: I'm a bit confused by Fig. 6. Why are there a different number of data points in panels a-d? Maybe this could be cleared up with a comment in the text or the caption of

Fig. 6. Are some data points "invisible" due to several being at the same location?

-L162: "The lower the chosen quantile, the clearer the scaling regime appears." Is this supposed to mean that lower quantiles show a smoother curve rather than the 3-regime form?

-L163: You've repeated "0.9999" here.

-L164: "The lower the quantile, the sparser the location of the quantile rainfall occurrence ..." I'm confused by this sentence. "Sparse" means "not dense". How can a location be sparse? Are you trying to say that for lower quantile events, the location of the maxima (Fig. 6) are more spread out across Germany?

-L169: "... the curve shows a very smooth ...". This is the curve in Fig. 5? I suggest writing the sentence as "... the curve (Fig. X) shows a very smooth ..." This makes it easier to follow for people who are reading the manuscript for the first time.

-L177: "The influence of ... persists until *the* hourly timescale ..."

-L188: "This result also implies the real rainfall process significantly deviates from the assumptions of the simple rainfall models suggested by Galmarini et al. (2004) and Zhang et al. (2013)." Would another possibility be that the 16-year time series used here is too short to see smooth behaviour at the point scale? I don't know how long the time series in the cited literature are.

-L182: I think here you mean "unprecedented" instead of "precedent".

-L193: "The maximum dept–duration relationships in Fig. 8 were clustered since some show a similar shape with each other." Did you really perform a k-Means clustering based on the 15 data points of Fig. 8, as suggested by the text here? This would be highly non-robust. I presume you really did the clustering based on all data points over DE, right? If so, please make this clearer in the text.

-L194: "The k-mean clustering successfully classified the depth–duration relationship into six categories ..." The word "successfully" here is a bit problematic without an objective method for deciding what "successful" is. K-Means will always categorize the data into the chosen number of classes, even if the data are completely unrelated to each other. If "successful" is based on the appearance of the right-hand panel of Fig. 9, a critic could say that to the naked eye there's little discernible difference between categories 3 and 4, or 2 and 6. You could just delete the word "successfully".

**References**

K. Lengfeld, P.-E. Kirstetter, H. J. Fowler, J. Yu, A. Becker, Z. Flamig, and J. J. Gourley. Use of radar data for characterizing extreme precipitation at fine scales

and short durations. *Environmental Research Letters*, 15(8):085003, 2020. doi: 10.1088/1748-9326/ab98b4.

---

## Referee Comment (RC3) · Anonymous Referee #3 · 2 Nov 2020

General comments:

The contribution provides an interesting information about maximum rainfall and its scaling with duration for Germany. The methodology is quite clear and plausible. The manuscript is well written and concise. There are however some points which need clarification and improvement (see detailed comments).

Detailed comments:

1. Line 73: Compared to other Countries in Germany PMP is not used directly for design. A brief comment about this would be useful.

[Figure]

2. Line 92: A brief summarization for the pre-processing of the radar data would be very useful, so that the reader has not to consult other papers (see also comment 5).

3. Eq. (6): A closing bracket is missing before the equal sign.

4. Eq. (7): The second factor of the right side of the equation should read "tau to the power of b".

5. Line 140ff: The study is analysing maximum observed values from radar data. Consider-ing the problem of clutter in radar data I am wondering that this seems not to influence the analysis very much, since these values "simulate" high precipitation intensities. Even if in radar pre-processing the clutter have been removed, there are usually still some of those left. Please, discuss this problem.

6. Line 43: "(blue solid line)" I don't see a solid line in Fig. 4.

7. Figure 4: It would be interesting to see the maximum observed values from rain gauges for the same period as the radar data. This would in comparison with the longer period al-low a discussion about sample size and record length (space vs time).

8. Lin 159: A quantile is one value, so it should read for instance "0.99999 is the forth great-est cell value" not the plural "... cells".

9. Figure 6: This figure does not make sense to me. It shows the locations of different quan-tile values. However, it would make more sense to show all values which exceed the prob-abilities and not only the one exact quantile.

10. Line 164ff: "The lower the quantile, the sparser the location ..." I don't understand this. From my point of view, I would say "The lower the quantile the more cells occur exceeding this quantile."

11. Figure 7: I would suggest scaling the colours not simply from minimum to maximum (may be from 0.1 to 0.9 quantile and non-linear), so that we have more contrast and not only blue in the figures.

12. Figure 8: What is the reason for selecting cities here? There might even be anthropogenic influence on rainfall in urban areas. Please discuss.

13. Figure 10: Please include description of the lines in caption or legend.

---

## Referee Comment (RC4) · Anonymous Referee #4 · 3 Nov 2020

In this paper radar derived Quantitative Precipitation Estimates with high temporal and spatial resolution are used to derive depth-duration relation for Germany. The result indicates that the scaling behaviour between the maximum rainfall depth and duration curves don't follow a power law function as previously derived by historical records. Instead, three distinct scaling regimes are identified which boundaries are 1h and 1d. The results are shown for different quantile levels and cities in Germany. Moreover the maximum rainfall depth-relation curves are derived for all radar pixels and clustered according to their shapes. This gave a presentation of the spatial relations and the different rainfall event type occurring over each pixel. The topic is very interesting and relevant and justifies a publication, however the manuscript suffers from several issues

that need to be addressed/discussed.

Major comments

In the introduction, a lot of focus is put on PMP estimation. This is part of the story, however, other aspects related to this topic should be considered and discussed here as well, e.g. rainfall extremes, the problems associated with radar QPE, rainfall extremes as not being a point event but rather a space-time phenomena, scaling properties of extremes, trading space for time, etc. Furthermore, I would expect the extremes detected by radar to look differently depending on the distance from the radar and the height above ground since the size of the radar bins increase with increasing distance and so does the elevation above ground. Thus, I would expect less severe extremes towards the outer areas of the radar circles. For example, many of the 5 min extremes in Figure 7 seem to be located near the sites of the radars. Last but not least, even though data correction was applied by the DWD, there is still uncertainty in the observed data, especially for extreme events. This should be mentioned since the results are derived from this product

I'm not convinced by the clustering applied for the scaling behaviour. The number of 6 clusters seems arbitrary and section 2.2.3 is poorly written, also with respect to the missing values. The k-means does not provide any measure for the quality of the classification. This all has implications on the results discussed in Section 3.5. How would the results look like if you chose 4 or 7 clusters? Did you perform a sensitivity analysis on how the results change of the number of clusters is changed? Could e.g. a fuzzy-logic based algorithm maybe yield better results?

Furthermore, I do not understand the concept behind Figure 8b. It shows the maximum difference before and after a gap, but what does this mean if >50% data are missing as e.g. in the northernmost part in Germany? Could you explain this more clearly? Why are the values generally higher the closer they are to the radar site? Could this also point to the different behaviour of extremes depending on the distance from the radar?

Minor comments

In general, the manuscript should be proof-read again, there a many awkward formulations, spelling mistakes, etc.

Specific Comments:

P1L17ff.: This whole sentence sounds weird, fatal disasters don't react to anything

P1L20: Introduce the acronym PMP here

P2L24: PMP can be estimated

P2L52: AR(1) -> first-order autoregressive process?

P3L58: Breña-Naranjo (this needs to be corrected in the references as well)

P3L63: 16 years

P4L77: Aren't there currently 17 C-Band Radars?

P4L78: delete "free and purchasable"

P4L81: ground information

P4L90f.: I find the justification that "Due to comparison reasons with another study at our institute only years 2001 to 2016 had been used for this study" rather weak. It would have been worthwhile to use the data until 2018, since you also mention that "With longer available time series of radar in the future, the deviation can be further investigated and tested" in the discussion.

P4L95-100: there is no need to mention the data size or how the data was saved.

P4L99: Why don't you use "NaN" for missing values?

P5L103-105: Are data of overlapping radar coverage areas similar? Since the data was measured by two different radars, the values can differ significantly (e.g. Yan and Bárdossy, 2019)

[Figure]

P5L107ff.: This whole paragraph is difficult to read, please rewrite this in a clearer way.

P6L115: How was the aggregation done considering the missing values? And how where events separated? Did you use a threshold? If yes, which?

P6L116: Durations of up to 3 h or 3 d? This whole sentence is difficult to read.

P6L130: Is the scaling relationship formulas are not correct

P7L138: Please reformulate sentence (also see major comments above)

P7L143: The temporal resolution of the ground truth reference should be mentioned. Furthermore, "world record" should be used (also in caption of Fig. 4)

P8L149: what are "very distant places of Germany"? Distant from what?

P8L151f.: Which temporal resolutions did you use for your analysis? 5 Min increments up to 16h? 3 days? Please specify!

P8L156: Data uncertainty is mentioned here but it's effect is not discussed!

P8L163: the first two quantiles are identical

P8L167: development the rainfall-duration → development of the rainfall-duration

P8L170f.: The statement that extreme rainfall events share common characteristics such as peak rainfall depth and correlation structure regardless of time-scale is a 'strong' statement that somehow contradicts the fact the rainfall extreme are spatially and temporally variant and their correlation structure differs.

Figure 6: A discrete colour bar should be used, moreover the spacing between the durations does not reflect the real spacing. An additional suggestion would be to add a second colour bar showing the associated rainfall values. This can help relate the quantiles to the rainfall values.

Figure 6: Why are the 0.99 quantiles are mostly located in the south of Bayern?

Figure 7: please redo with a discrete colour bar and maybe scale it to the values so the details in the map are more visible.

P12L180: avoid formulations such as "really high"

P12L185: Stuttgart

P12L187: who

P12L188 what do you mean with "real rainfall process"?

Figure 8: How are curves for the cities calculated, mean of all cells in the city or maximum cell? This might be relevant to explain why neighbouring cities show very distinct behaviour. P13L193ff. Rewrite this paragraph and be more specific, what is "successfully classified", what is a "certain colour", etc.

P13L193: dept-duration → depth-duration

P13L199: "that pour for around 1 h and move on or weaken" -> rewrite this

Figure 9: Does this relate to topography?

Figure 9: Legend of plot 'mm/uration' → mm/duration

Figure 9: How would the clustering and this map look, if the data was divided between summer and winter period? Did you look into this?

P15L204f.: If the look similar and occur together why do you distinguish these categories? (c.f. major comments above)

P15L205: a slope is steeper instead if higher

P15L213: the term 'super-daily' is confusing, please consider changing it.

P15L219: saying that areas with category 5 have never been hit by any 'extreme' extreme event needs more evidence. It could be that the occurred events were not well captured due to data uncertainty.

[Figure]

P15L232: areas don't "experience" a rainstorm...

P16L232: ... the same goes for pixels!

P16L264f.: Reformulate this sentence

Figure 10: Please add legend and increase the grid visibility.

P17L268f. If you have the data until 2018, why didn't you use them? (c.f. P4L90f.)

References: Several issues with capitalization of titles and author's names.

---

## Author Comment (AC2) · 8 Dec 2020

**nhess-2020-1982: An analysis on temporal scaling behaviour of extreme rainfall of Germany based on radar precipitation QPE data (Pöschmann et al.)**

**Reply to the comments from Referee #2:**

We appreciate the unbiased review and are thankful for all comments and suggestions from anonymous referee #2 that helped to improve our manuscript. We have provided answers to all comments in blue and updated the original manuscript accordingly. All line numbers given in our responses refer to the original version of the manuscript.

**Comments:**

**General Comments**

The authors present an analysis of precipitation intensity-duration relationships over Germany based on the RADKLIM radar dataset, with a spatial resolution of 1 km² and a temporal resolution of 5 minutes. They find a non-smooth scaling relationship, with indications of regime transitions between different temporal aggregation lengths. I think the study is interesting and I'm not aware of a similar study using radar data over Germany. I didn't see any fundamental flaws in the work and therefore think it could be publishable.

Before it can be published though, I think there is room for improvement. This mostly relates to the text and presentation, rather than the science. In some parts the manuscript can be a bit confusing and hard to follow, with aspects explained in a sub-optimal manner. I've outlined my suggestions below.

I should note that, in order to perform an independent and unbiased review, I refrained from reading the already published reviewer comment and therefore apologise for any repetition of what may have already been said!

**Main Comments**

**1. The "three-regime" scaling curve**. It wasn't clear to me if the "three-regime" scaling curve you report is a new finding or not, i.e. is there any other literature which report a multiple-regime scaling curve? If the three-regime scaling curve is a novel result, then you should emphasize this. If it is not, then you should cite other studies where multiple-regime scaling curves were reported.

**Response:** We replied together with your main comment 3.

**2. The RADKLIM data set.**

(a) As the whole study and its results hinge on the RADKLIM dataset, I think the reader needs to be given more information about this dataset and its limitations, particularly how they might affect the results of an extreme precipitation study. This is particularly important because there is no documentation for the RADKLIM dataset available in English. As far as I know, the only available source is DWD Report No. 251, which is in German (https://www.dwd.de/DE/leistungen/pbfb_verlag_berichte/pdf_einzelbaende/251_pdf).

For example, RADAR data are known to often contain artefacts due to interference (wind turbines, WLAN networks, etc.), which can be particularly problematic when looking at intense events. What

steps were taken in the production of RADKLIM to eliminate or reduce such artefacts? You'll find all the necessary information in Section 4 of DWD Report No. 251 (see link). Of course we can't expect the data and results to be perfect, so we need to transparently present these issues to the reader to help them form their own opinions.

**Response:** The publication of Kreklow et al. (2019) shows that the remaining weaknesses of RADKLIM are a higher number of missing values as well as an overall negative bias causing an underestimation of high intensity rainfall due to spatial averaging and rainfall-induced attenuation of the radar beam. The authors assume that almost half a million pixels will still provide an adequate representation of the characteristics and feel safe that there is no danger of getting unreasonably high values for the maxima.

Lines 81 – 84 were edited as follows:

"Since the quality enhancement of RADOLAN is ongoing without post-correcting previous data, the so–called radar climatology project of the DWD, RADolanKLIMatologie (RADKLIM,Winterrath et al., 2017) has consistently reanalysed the complete radar data archive set since 2001 for improved homogeneity despite the originally different processing algorithms. Compared to RADOLAN, RADKLIM has implemented additional algorithms leading to consistently fewer radar artefacts, improved representation of orography as well as efficient correction of range-dependent path-integrated attenuation at longer time scales (Kreklow et al., 2019). Whereas RADOLAN is not well suited for climatological applications with aggregated precipitation statistics, RADKLIM is a promising data set for these climatological applications. The RADKLIM data is available.."

Lines 91 – 94 were edited as follows:

"The YW product covers the area composed of 1100 x 900 pixels with the spatial resolution of 1 km (improved compared to former version of RADOLAN). Remaining weaknesses of RADKLIM (as outlined in Kreklow et al. (2019)) are the greater number of missing values (compared below) compared to RADOLAN as well as negative bias causing an underestimation of high intensity rainfall due to spatial averaging and rainfall-induced attenuation of the radar beam."

The following sentence was added at line 151 (centre):

"As mentioned in the data quality description, it is possible that these sub-hourly values do not represent the true extreme across Germany for 2001-2016 since radar-based measurements at fine timescale (e.g. xx minutes) are highly sensitive to the averaging effects."

Following sentence was added at the end of the conclusion:

"Also, the known issue of rainfall extreme underestimation by RADKLIM-YW and the potential impact on the results need further investigation."

(b) The scaling regime transitions at 1 hour and 1 day got me thinking. As we know, RADKLIM uses station data to adjust radar-measured precipitation. According to DWD Report No. 251 (Sections 4c and 4d), this is done with hourly station data where available. If no hourly station data are available, then daily station data are used. The number of hourly and daily stations used can be seen in Fig. 5 of the aforementioned report. The radar data are summed to the temporal resolution of the station data and adjusted, before a "disaggregation" procedure is applied to return the radar data to their original temporal frequency.

Could it be that the different regimes result from this adjustment process? Maybe those pixels adjusted with hourly gauge data tend towards a scaling curve with one characteristic slope, while those pixels adjusted with daily gauge data tend towards a scaling curve with a different characteristic slope? What do the authors think? I think this question underlines the importance of my first main point about whether your multiple-regime scaling curve is a unique finding or not. If a multiple-regime scaling curve has never been reported before (even in radar-based studies), then it would arouse concern that your three-regime scaling curve may be a data artefact. If multiple-regime scaling curves are common, then my comment can likely be ignored.

**Response:** Thank you for your interesting comment, this has previously been a topic of discussions between the authors. However, we did not agree after all, mainly because Figure 9 would in our opinion show different curve characteristics than it does.

We furthermore hesitate to emphasize on the multiple-regime scaling curve as a novel finding, since it most likely simply results from the lengths of the radar time series. The lack of time length cannot be compensated by more information in space to the extent we hoped. Obviously, some maxima are captured that would not be captured without the detail of spatial information, however for a single scaling regime additional big contributing events and maxima would be needed, which seems only possible with more years of data.

Additionally, for example the study of Galmarini et al. 2004 proposes a simple rainfall model, however, their curves of real data (they evaluated station-wise) in the beginning of the paper in fact show some similar deviations from a straight line. This is similar to what you see when looking at just one pixel in our data set.

One finding of our paper is, however, as mentioned in the discussion, that it might be possible that the regional or world record curves which are often based on fixed interval sampling and "limited" to rain gauge locations COULD potentially look different (= without the clear scaling regime) if more spatial information is considered. Especially when looking at Figure 10, we could show that even a sample size of 1000 pixels can give a very wrong representation of the "real" information. This is similarly true for rain gauge data, since the gauges themselves already represent an exceedingly small unit.

We thus hope that our study opens more discussion on that topic, but we would refrain from making statements that we after all are not able to fully prove (for now).

**3. Figure Captions**. I think it would help the readers if the figure captions were a bit more descriptive. Generally, they are just one sentence. For example, in the captions you could add some more text highlighting the interesting aspects of the figures, so that it is clear to the reader what exactly the motivation for showing the plot is and why the presented result is interesting. This saves the reader from having to flip back and forth between the text and image (which may be several pages apart).

**Response:** We edited the captions of the following figures as follows:

Figure 2: Results of NaN analyses of the QPE RADKLIM-YW from 2001–2016: (a) Spatial distribution of the proportion of NaNs (in %) for each pixel, (b) Maximum intensity per time step of 5 min that need to be interpolated (= maximum intensity difference within one time step to overcome)

(Figure 3: removed from manuscript)

Figure 4: Overview of maximum rainfall records in Germany. Chart: Depth-duration relationship of rainfall records based on QPE RADKLIM-YW (data of this study) (blue dots), and as reference the relationships based on the German ground network Rudolf and Rapp, 2003; DWA, 2015; DWD, 2020) (non-filled triangles) and the "world records" (World Meteorological Organization, 1994; NWS, 2014). Map: Locations of rainfall maxima (based on QPE RADKLIM-YW) for the considered duration.

(Note: some studies had been updated due to a suggestion of referee#1)

Figure 5: Depth-duration relationships of rainfall values of whole Germany based on QPE RADKLIM-YW for 2001 – 2016 from maximum values down to the 3921$^{st}$ greatest per duration

Figure 6: Locations of the 0.99999, 0.9999, 0.999, and 0.99 quantile rainfall with varying durations from 5 min to 3 d. Point colours represent the corresponding rainfall duration, similar for each quantile. Different numbers of data points in panels a-d result from several data points being at the same location.

Figure 7: Spatial distribution of the maximum rainfall values retrieved from QPE RADKLIM-YW (2001 – 2016) for different durations (5 min to 3 days).

Figure 8: Depth-duration relationships of rain records for single pixels at rain gauge locations within state capitals of German federal states.

Figure 9: Resulting 6 groups after clustering the maximum depth- duration relationships of rainfall for all pixels. The left panel shows the spatial distribution of the groups, distinguishable by colour. The corresponding curve shapes of 100 randomly selected radar pixels from each group are displayed on the right side with the same colours as the map.

Figure 10: Dependency of maximum depth-duration relationship characteristic on underlying pixel sample size. The maximum rainfall values are derived from (a) 10, (b) 100, (c) 1000, and (d) 10000 random pixels from all considered pixels (n=392 128) within Germany. For each sample size, 30 ensembles are displayed and compared to the overall maximum curve from Fig. 4 and 5 (yellow top line in (a) – (d)).

**Minor Comments and Technical Corrections**

-Language: As far as I know, NHESS publishes using British English. The manuscript currently uses American English spellings. If you change the language of your spell-checker to British English you should easily be able to find all of these misspellings. For example, "behavior → behaviour", "modeled → modelled", "color → colour", "neighboring → neighbouring", etc.

**Response:** Thank you for this comment! We have change it accordingly.

-Superfluous text: I think there's a fair bit of superfluous text which could be eliminated. As just a small example, I don't think anyone is interested that (L78) the DWD "is providing different free and purchasable rainfall data derived from it". Maybe I'm being a bit picky here, so you can ignore my comment if you want! Less redundant text is, in general, always appreciated by the reader.

**Response:** Thank you for this remark. We will try to eliminate it.

-Title: "An analysis *of* temporal scaling behaviour of extreme rainfall *in* Germany based on radar precipitation QPE data" (not `on' or `of')

**Response:** Thank you! We will request a change of the title!

-L38: Here you've switched from "d" to "D".

**Response:** We have changed d to D for consistency reasons and also edited lines 33 ff as follows (based on the first referee's comments):

"Jennings discovered that this unique scaling behavior holds at the rainfall duration between 1 min through 24 months. Paulhus (1965) showed that the same power law relationship holds after addition of new world rainfall record observed at the island of La Réunion at the duration between 9 h and 8 d. The envelope for this extreme values can be expressed as: $P = \alpha D^{\beta}$ (1) where P is the maximum precipitation (in mm) occurring in duration D (in h), the coefficient α (425 in Paulhus (1965)) represents the value at one hour of the depth–duration relationship plotted on the log–log plane, and the exponent β (0.47 in Paulhus (1965)) is the parameter characterizing the scaling behavior of the depth–duration relationship. The Spanish study of Gonzales and Bech (2017) updated the world envelope's slope to 0.51, showing a remarkable stability. Multiple exponents describing the scaling property of ……."

-L55-: For your discussion of the impact of sparse rain gauge networks (also elsewhere in the manuscript), the very new publication of Lengfeld et al. (2020) based on the RADKLIM network may be particularly interesting for you.

**Response:** Thank you for suggesting this interesting publication. We have included it in the introduction.

-L88: What are RADOLAN-RY and RADOLAN-RH?

**Response:** We removed RADOLAN-RH in the text, since it is just the 1-hour sum of RADOLAN-RY. We added a short information in brackets: "… 5 min product RADOLAN-RY (rainfall estimate after basic quality correction and refined z-R-relationship) with the help of RADKLIM-RW …."

-Eq. 7: Something has gone wrong here. If log(M) = B + b*τ, then M = 10(B+b*τ). Also, there's an open bracket in Eq. 6.

**Response:** We merged sections 2.2.1 and 2.2.2 into one paragraph 2.2. "Depth-Duration relationships" and removed most of the equations for a better reading.

The methodology section is now shortened as follows:

"2.2 Depth-Duration relationships
Maximum rainfall values for each duration τ between 2001-2016 were calculated with rolling sums applied over moving windows using the R package RcppRoll (Ushey, 2018). Durations of up to 3 d were chosen for the analysis, with multiple steps for minutes and hours out of our interest for sub–hourly and sub–daily pattern. The records may include non-rainfall data and thus do not imply continuous precipitation for the period considered. Values were not aggregated spatially, since this usually reduces the maximum intensity values (Cristiano et al., 2018). First, the extreme values for each pixel and duration $M_{max}^{\tau,pixel}$ are calculated. Afterwards, the overall maxima for whole Germany for each τ ($M_{max}^{\tau}$) is extracted from these calculated extreme values. Based on these results, the depth–duration relationships can be built for each pixel as well as for the whole of Germany."

Section 2.2.3 is thus changed to 2.3.

-L121: Please state how many pixels exist for the whole of Germany, i.e. N = ? This is also useful to know when we look at the subsamples in Fig. 10. I therefore also suggest repeating the value of N in the caption of Fig. 10 (like you show in Fig. 5) and also somewhere around the line 235-238.

**Response:** The whole of Germany has around 392875 pixels (uncertainty involved by extracting the data with the polygon of Germany). We evaluated those N = 392128 pixels (compare Figure 2) covered by the German radar data. We left all pixels with data in the analysis, knowingly that some (grey colour in Figure 2) have shorter time series than 16 years.

We added the total number of analysed pixels in line 85: "The RADKLIM data is available in the following two formats, with around 392 128 filled pixels within the German border" and in line 238. Additionally, we repeated the value in caption of Fig. 10 (compare related comment with new figure captions).

-L138: There's an open bracket here too.

-Fig. 3: Personally I think that Fig. 3 is superfluous. You could just state the result in one sentence without showing the plot, and take up less space. k-Means and elbow plots are pretty standard and are unlikely to confuse readers. Do you know that the journal publication charge will be based on the number of pages? Alternatively, if you really like the plot you could put it in a supplementary info file or an appendix.

**Response:** Referee #1 asked us to give a more detailed explanation of the classification. We still followed your suggestion and removed the Figure 3. The text edit of section 2.2.3 (new: section 2.3) is below:

"The depth-duration relationships ($M_{max}^{\tau,pixel}$ vs $\tau$) for each pixel derived from Sect. 2.2 are individually clustered with the K-Mean clustering algorithm (Scott and Knott, 1974). "Erroneous" pixels (=having NaNs as resulting maxima) were excluded from the cluster process in order to avoid disturbances. The data was rescaled to make the characteristics more comparable with each other. If the number of clusters is not predefined, it can be identified by drawing an elbow chart. For different numbers of clusters K the measure of the variability of the observations within each cluster (Total within-cluster sum of squares, y-axis) is calculated and the curve should bend like an elbow at the optimal value. Since the algorithm did not suggest a number of clusters, we chose six clusters for a sufficiently detailed analysis since it gave consistent results when repeating the automatic algorithm for several times (each time the algorithm clusters slightly differently)." [end of section]

-L143 and Fig. 4: I can't see any "blue solid lines" or "red dotted lines" in Fig. 4! I only see blue dots, black triangles and empty triangles. Also, shouldn't the unit in Fig. 4 y-axis just be mm? The caption for Fig. 4 is also confusing, because it talks about "Spanish ground gauge records" but these aren't visible in the plot.

**Response:** We changed it to the characteristics shown in Figure 4: Triangles (filled/empty) and dots. The Spanish study indeed is not included in the Figure, so we removed it in the caption.

-Fig. 6: I'm a bit confused by Fig. 6. Why are there a different number of data points in panels a-d? Maybe this could be cleared up with a comment in the text or the caption of Fig. 6. Are some data points "invisible" due to several being at the same location?

**Response:** Indeed, some data points are "invisible" due to several being at the same location! We added a comment in the caption (see new figure captions above)

-L163: You've repeated "0.9999" here.

**Response:** We did not notice, thanks a lot! We changed the first one to 0.99999.

-L162: "The lower the chosen quantile, the clearer the scaling regime appears." Is this supposed to mean that lower quantiles show a smoother curve rather than the 3-regime form?

**Response:** Yes! We changed the sentence to "Lower quantiles thus show a smoother curve rather than the 3-regime form."

-L164: "The lower the quantile, the sparser the location of the quantile rainfall occurrence ..." I'm confused by this sentence. "Sparse" means "not dense". How can a location be sparse? Are you trying to say that for lower quantile events, the location of the maxima (Fig. 6) are more spread out across Germany? **(s. below)**

-L169: "... the curve shows a very smooth ...". This is the curve in Fig. 5? I suggest writing the sentence as "... the curve (Fig. X) shows a very smooth ..." This makes it easier to follow for people who are reading the manuscript for the first time.

**Response:** find the text edition from line 164 (including referee #1's suggestions/comments):

"It shows that the number of locations increases the lower the quantile of maximum rainfall is. This suggests the reduction of the influence of one single rainfall event on the depth-duration relationship causing inflection in the curve. Additionally, from a certain degree of quantile (Fig. 6 d) the locations of maximum rainfall contributing to the development of the rainfall-duration relationship seem to happen mainly in the wider Alpine region in South Germany. This suggests that rather natural rainfall mechanisms are dominating the scaling relationship, such as regional characteristics and meteorological conditions (e.g. orographic lifting or leeward effects). Naturally, one would assume that this heterogeneity of the meteorological conditions and rainfall generating mechanisms will reflect rather regional characteristics and will exhibit some irregular scaling behavior. Contrary to this conjecture, the curves in Fig. 5 (99.9% and 99%) show a rather smooth scaling behavior" [end of subsection 3.2]

-L177: "The influence of . . . persists until *the* hourly timescale ..."

**Response:** Thank you! We changed it!

-L188: "This result also implies the real rainfall process significantly deviates from the assumptions of the simple rainfall models suggested by Galmarini et al. (2004) and Zhang et al. (2013)." Would another possibility be that the 16-year time series used here is too short to see smooth behaviour at the point scale? I don't know how long the time series in the cited literature are.

**Response:** Galmarini et al. (2004) uses time series of different lengths and resolutions, from 1 year of 1 min resolution up to 160 years with 1day resolution. Zhang 2013 does actually not provide any information on the time length and focuses on 1 time series. Figure 3 in Galmarini et al. (2004) shows that longer time series do not necessarily show smoother scaling behaviour. The 6 min/99 years curve for example shows multiple scaling (also with bends at 1 hour/1 day), and there seems to be no clear dependency on time length or resolution.

We changed the wording of the passage after reading your comment, to: [replacing l.188 ff]

"Furthermore, Galmarini et al. (2004) and Zhang et al. (2013) both showed that the maximum rainfall-duration relationship at a given point location follows a smooth and simple power law if

the rainfall process can be modelled with a set of simple stochastic processes. Our results imply that natural rainfall processes might significantly deviate from this rather simple assumption, also their model frameworks are based on very few time series of very different lengths and resolutions.

-L182: I think here you mean "unprecedented" instead of "precedent".

**Response:** We mean that the 2002 flooding in Saxony had long heavy rainfall before the event. But we agree, that "unprecedented" would also be a good term to use here thus we changed it accordingly.

-L193: "The maximum dept-duration relationships in Fig. 8 were clustered since some show a similar shape with each other." Did you really perform a k-Means clustering based on the 15 data points of Fig. 8, as suggested by the text here? This would be highly non-robust. I presume you really did the clustering based on all data points over DE, right? If so, please make this clearer in the text.

**Response:** We added a half-sentence that Figure 9 clustering is for all pixels as described in the corresponding section 2.2.3
Edition of first sentence in line 193:
"The maximum depth–duration relationships for all pixels within Germany were clustered since Fig. 8 indicated that they might show similar shapes."

Additionally, we mentioned in Figure 9 also (compare new figure caption) that the clustering is based on all pixels in Germany.

-L194: "The k-mean clustering successfully classified the depth-duration relationship into six categories ..." The word "successfully" here is a bit problematic without an objective method for deciding what "successful" is. K-Means will always categorize the data into the chosen number of classes, even if the data are completely unrelated to each other. If "successful" is based on the appearance of the right-hand panel of Fig. 9, a critic could say that to the naked eye there's little discernible difference between categories 3 and 4, or 2 and 6. You could just delete the word "successfully".

**Response:** We deleted it!

References

K. Lengfeld, P.-E. Kirstetter, H. J. Fowler, J. Yu, A. Becker, Z. Flamig, and J. J. Gourley. Use of radar data for characterizing extreme precipitation at _ne scales and short durations. Environmental Research Letters, 15(8):085003, 2020. doi: 10.1088/1748-9326/ab98b4.

Reference

J. Kreklow, B. Tetzlaff, G. Kuhnt, and B. Burkhard. A Rainfall Data Intercomparison Dataset of RADKLIM, RADOLAN, and Rain Gauge Data for Germany, Data, 4, 118, https://doi.org/10.3390/data4030118, 2019.

---

## Author Comment (AC3) · 8 Dec 2020

*nhess-2020-1982:* An analysis on temporal scaling behaviour of extreme rainfall of Germany based on radar precipitation QPE data (Pöschmann et al.)

**Reply to the comments from Referee #3:**

We are thankful for comments and suggestions from the anonymous referee #3 that helped us to improve our original draft. We have provided responses to all comments in blue and updated the manuscript accordingly. All line numbers given in our responses refer to the original version of the manuscript.

**Comments:**

**General comments:**

The contribution provides an interesting information about maximum rainfall and its scaling with duration for Germany. The methodology is quite clear and plausible. The manuscript is well written and concise. There are however some points which need clarification and improvement (see detailed comments).

**Detailed comments:**

1.  Line 73: Compared to other Countries in Germany PMP is not used directly for design. A brief comment about this would be useful.

**Response:** Thank you for your comment! In order to avoid more text, we chose to simply remove the two last sentences from the introduction from the manuscript.

2.  Line 92: A brief summarization for the pre-processing of the radar data would be very useful, so that the reader has not to consult other papers (see also comment 5).

**Response:** s. our text edits listed in Comment 5

3.  Eq. (6): A closing bracket is missing before the equal sign. (answer together with next point)

4.  Eq. (7): The second factor of the right side of the equation should read "tau to the power of b".

**Response:** Following Reviewer#1 we have restructured the section and removed all equations.

We merged sections 2.2.1 and 2.2.2 into one paragraph 2.2. "Depth-Duration-Relationships" as follows:

"2.2 Depth-Duration relationships
Maximum rainfall values for each duration τ between 2001-2016 were calculated with rolling sums applied over moving windows using the R package RcppRoll (Ushey, 2018). Durations of up to 3 d were chosen for the analysis, with multiple steps for minutes and hours out of our interest for sub–hourly and sub–daily pattern. The records may include non-rainfall data and thus do not imply continuous precipitation for the period considered. Values were not aggregated spatially, since this usually reduces the maximum intensity values (Cristiano et al., 2018). First, the extreme values for each pixel and duration $M_{max}^{\tau,pixel}$ are calculated. Afterwards, the overall maxima for whole Germany for each τ ($M_{max}^{\tau}$) is extracted from these calculated extreme values. Based on these results, the depth–duration relationships can be built for each pixel as well as for the whole of Germany."

5.  Line 140ff: The study is analysing maximum observed values from radar data. Considering the problem of clutter in radar data I am wondering that this seems not to influence the analysis very much, since these values "simulate" high precipitation intensities. Even if in radar pre-processing the clutter have been removed, there are usually still some of those left. Please, discuss this problem.

**Response:** After having worked with RADOLAN, there was some doubt concerning the quality of the RADKLIM product. We expected radar artefacts that would destroy our results, similar to what you mentioned. However,

we did not find any obvious problematic "high" precipitation intensities and were confirmed by Kreklow et al. (2019) who explained that remaining weaknesses of RADKLIM are a higher number of missing values as well as an overall negative bias, causing a rather "underestimation" of high intensity rainfall due to spatial averaging and rainfall-induced attenuation of the radar beam.

We agree that a little more text on the data quality would add to a better understanding and thus edited several passages in the document as follows:

Lines 81 – 84 were edited as follows:

"Since the quality enhancement of RADOLAN is ongoing without post-correcting previous data, the so–called radar climatology project of the DWD, RADolanKLIMatologie (RADKLIM,Winterrath et al., 2017) has consistently reanalysed the complete radar data archive set since 2001 for improved homogeneity despite the originally different processing algorithms. Compared to RADOLAN, RADKLIM has implemented additional algorithms leading to consistently fewer radar artefacts, improved representation of orography as well as efficient correction of range-dependent path-integrated attenuation at longer time scales (Kreklow et al., 2019). Whereas RADOLAN is not well suited for climatological applications with aggregated precipitation statistics, RADKLIM is a promising data set for these climatological applications. The RADKLIM data is available.."

Lines 91 – 94 were edited as follows:

"The YW product covers the area composed of 1100 x 900 pixels with the spatial resolution of 1 km (improved compared to former version of RADOLAN). Remaining weaknesses of RADKLIM (as outlined in Kreklow et al. (2019)) are the greater number of missing values (compared below) compared to RADOLAN as well as negative bias causing an underestimation of high intensity rainfall due to spatial averaging and rainfall-induced attenuation of the radar beam."

The following sentence was added at line 151 (centre):
"As mentioned in the data quality description, it is possible that these sub-hourly values do not represent the true extreme across Germany for 2001-2016 since radar-based measurements at fine timescale (e.g. xx minutes) are highly sensitive to the averaging effects."

Following sentence was added at the end of the conclusion:

"Also, the known issue of rainfall extreme underestimation by RADKLIM-YW and the potential impact on the results need further investigation."

6. Line 143: "(blue solid line)" I don't see a solid line in Fig. 4.

**Response:** We changed it to the characteristics shown in Figure 4: Triangles (filled/empty) and dots.

7. Figure 4: It would be interesting to see the maximum observed values from rain gauges for the same period as the radar data. This would in comparison with the longer period allow a discussion about sample size and record length (space vs time).

**Response:** Thank you for this suggestion! However, we chose to not add another uncertainty into the analysis. The gauge data used for Figure 4 is available as final product and we did not calculate it ourselves, and a time series with fine resolution of 5 minutes is not freely available, which would make the gauge data more comparable to our data. There exist data with 10 minutes resolution, but the quality of the data is not verified.

8. Lin 159: A quantile is one value, so it should read for instance "0.99999 is the forth greatest cell value" not the plural ". . . cells".

**Response:** This is true, we change it!

9. Figure 6: This figure does not make sense to me. It shows the locations of different quantile values. However, it would make more sense to show all values which exceed the probabilities and not only the one exact quantile.

**Response:** We are sorry that the interpretation of the Figure is unnecessarily difficult. The purpose of placing the Figure like this is related to Figure 5: We want to show that if not taking the maxima of

maxima, but certain quantiles of maxima (we chose to take numbers that correspond to 99.999%, 99.99 %, 99.9% and 99%, but obviously other values could have been chosen):

1) Seven locations (as in Figure 4) no longer hold all maxima, but the number of locations is increased and locations are also spread over all of Germany.

2) Interestingly, despite 1), the corresponding depth-duration relationships are straightening out (=getting smoother) and start to reflect rather natural rainfall conditions (e.g. dominance of Alpine region in Figure 6d) instead of seen before singular "extremes of extremes".

10. Line 164ff: "The lower the quantile, the sparser the location . . ." I don't understand this. From my point of view, I would say "The lower the quantile the more cells occur exceeding this quantile."

**Response:** Your answer is true, even though we want to focus on a different aspect. Find our text edition from line 164ff (including referee #1 and #2's comments):

"It shows that the number of locations increases the lower the quantile of maximum rainfall is. This suggests the reduction of the influence of one single rainfall event on the depth-duration relationship causing inflection in the curve. Additionally, from a certain degree of quantile (Fig. 6 d) the locations of maximum rainfall contributing to the development of the rainfall-duration relationship seem to happen mainly in the wider Alpine region in South Germany………"

11. Figure 7: I would suggest scaling the colours not simply from minimum to maximum (may be from 0.1 to 0.9 quantile and non-linear), so that we have more contrast and not only blue in the figures.

**Response:** If the referee would find it more suitable, we could change the colour scheme as given in the following example (colours represent quantiles):

[Figure]

12. Figure 8: What is the reason for selecting cities here? There might even be anthropogenic influence on rainfall in urban areas. Please discuss.

**Response:** Thank you for this comment! We chose these cities since they are distributed relatively well over Germany. We could have also selected random pixels, but preferred to choose known locations. The purpose of showing it is to reflect the variation in the depth-duration relationships. We believe that even anthropogenic influence will not change the message of the figure.

13. Figure 10: Please include description of the lines in caption or legend.

**Response:** We edited Figure 10's caption as follows:

"Dependency of maximum depth-duration relationship characteristic on underlying pixel sample size. The maximum rainfall values are derived from (a) 10, (b) 100, (c) 1000, and (d) 10000 random pixels from all considered pixels (n=392 128) within Germany. For each sample size, 30 ensembles are displayed and compared to the overall maximum curve from Fig. 4 and 5 (yellow top line in (a) – (d))."

**Reference**

J. Kreklow, B. Tetzlaff, G. Kuhnt, and B. Burkhard. A Rainfall Data Intercomparison Dataset of RADKLIM, RADOLAN, and Rain Gauge Data for Germany, Data, 4, 118, https://doi.org/10.3390/data4030118, 2019.

---

## Author Comment (AC4) · 8 Dec 2020

*nhess-2020-1982:* An analysis on temporal scaling behaviour of extreme rainfall of Germany based on radar precipitation QPE data (Pöschmann et al.)

**Reply to the comments from Referee #4:**

We are thankful for anonymous referee #4's comments and suggestions in order to improve our original draft. We have provided responses to all comments in blue and updated the manuscript according to the suggestions. All line numbers given in our responses refer to the original version of the manuscript.

**Comments:**

In this paper radar derived Quantitative Precipitation Estimates with high temporal and spatial resolution are used to derive depth-duration relation for Germany. The result indicates that the scaling behaviour between the maximum rainfall depth and duration curves don't follow a power law function as previously derived by historical records. Instead, three distinct scaling regimes are identified which boundaries are 1h and 1d. The results are shown for different quantile levels and cities in Germany. Moreover the maximum rainfall depth-relation curves are derived for all radar pixels and clustered according to their shapes. This gave a presentation of the spatial relations and the different rainfall event type occurring over each pixel. The topic is very interesting and relevant and justifies a publication, however the manuscript suffers from several issues that need to be addressed/discussed.

**Major comments**

-   In the introduction, a lot of focus is put on PMP estimation. This is part of the story, however, other aspects related to this topic should be considered and discussed here as well, e.g. rainfall extremes, the problems associated with radar QPE, rainfall extremes as not being a point event but rather a space-time phenomena, scaling properties of extremes, trading space for time, etc.

**Response:** Thank for this remark, we have added some changes to the introduction:

We have added some text in the first paragraph:

[revised manuscript text omitted]

- Furthermore, I would expect the extremes detected by radar to look differently depending on the distance from the radar and the height above ground since the size of the radar bins increase with increasing distance and so does the elevation above ground. Thus, I would expect less severe extremes towards the outer areas of the radar circles. For example, many of the 5 min extremes in Figure 7 seem to be located near the sites of the radars. Last but not least, even though data correction was applied by the DWD, there is still uncertainty in the observed data, especially for extreme events. This should be mentioned since the results are derived from this product.

**Response:** Thank you for this comment. As shown in Figure 1 below (different colour scheme than Figure 7 in the original manuscript), the radius-dependency could not be identified by our analysis. When starting from the finest resolution of 5 minutes (compare Figure below with different colour scheme than Figure 7), we do not see any accumulation of extremes located near the sites of the radar and it is also not the case the more we aggregate. One possible reason is that a principle of RADOLAN is to take the maximum value where radar circles overlap (=outer areas of the radar circles) which would lead to a rather accumulation of higher values in the outer areas (this is not the case in our results). Additionally, because of orography we don't expect a general dependency on distance from radar and height above ground within the data. As Kreklow et al., 2019 explained, remaining weaknesses of RADKLIM are a higher number of missing values as well as an overall negative bias causing a rather "underestimation" of high intensity rainfall due to spatial averaging and rainfall-induced attenuation of the radar beam. We generally think that with the huge number of pixels evaluated, the analysis still provides an adequate representation of the characteristics.
We agree that a little more text on the data quality would add to a better understanding and thus edited several passages in the documents as follows:

Lines 81 – 84 were edited as follows:

"Since the quality enhancement of RADOLAN is ongoing without post-correcting previous data, the so–called radar climatology project of the DWD, RADolanKLIMatologie (RADKLIM,Winterrath et al., 2017) has consistently reanalysed the complete radar data archive set since 2001 for improved homogeneity despite the originally different processing algorithms. Compared to RADOLAN, RADKLIM has implemented additional algorithms leading to consistently fewer radar artefacts, improved representation of orography as well as efficient correction of range-dependent path-integrated attenuation at longer time scales (Kreklow et al., 2019). Whereas RADOLAN is not well suited for climatological applications with aggregated precipitation statistics, RADKLIM is a promising data set for these climatological applications. The RADKLIM data is available.."

Lines 91 – 94 were edited as follows:

"The YW product covers the area composed of 1100 x 900 pixels with the spatial resolution of 1 km (improved compared to former version of RADOLAN). Remaining weaknesses of RADKLIM (as outlined in Kreklow et al. (2019)) are the greater number of missing values (compared below) compared to RADOLAN as well as negative bias causing an underestimation of high intensity rainfall due to spatial averaging and rainfall-induced attenuation of the radar beam."

The following sentence was added at line 151 (centre):
"As mentioned in the data quality description, it is possible that these sub-hourly values do not represent the true extreme across Germany for 2001-2016 since radar-based measurements at fine timescale (e.g. xx minutes) are highly sensitive to the averaging effects."

Following sentence was added at the end of the conclusion:

"Also, the known issue of rainfall extreme underestimation by RADKLIM-YW and the potential impact on the results need further investigation."

[Figure]

*FIG. 1: 5 Minutes Extremes for 2001 - 2016 derived from RADKLIM-YW*

- I'm not convinced by the clustering applied for the scaling behaviour. The number of 6 clusters seems arbitrary and section 2.2.3 is poorly written, also with respect to the missing values. The k-means does not provide any measure for the quality of the classification. This all has implications on the results discussed in Section 3.5. How would the results look like if you chose 4 or 7 clusters? Did you perform a sensitivity analysis on how the results change of the number of clusters is changed? Could e.g. a fuzzy-logic based algorithm maybe yield better results?

**Response:** Thank you for this comment. We agree with the reviewer in that our K-means clustering analysis may be subjective even though the number of the clusters chosen in this study (6) was the result of our careful visual analysis based on 2 through 8 clusters. However, we also would like to argue that the primary focus of this cluster analysis should not be the validity of the cluster numbers or the analysis methodology, but the fact that the scaling behaviour of the rainfall extremes can be classified based on the inflection points in the scaling relationship and this inflection points are primarily governed by the history of the extreme events that each of the radar pixels experienced. We believe that this analysis is particularly meaningful in that it specifically showed, from a spatial perspective, why the rainfall extreme value scaling behaviour deviates from a simple power law.

Since Referee#1 wanted a more detailed explanation, we have edited section 2.2.3 as follows (including an adaptation of the NaN part, but removed Fig. 3, since Ref#2 found it too superfluous). Please note that sections 2.2.1 and 2.2.2 were also summarized to 2.2, thus 2.2.3 will become section 2.3. in the revised paper:

"The depth-duration relationships ($M_{max}^{\tau,pixel}$ vs $\tau$) for each pixel derived from Sect. 2.2 are individually clustered with the K-Mean clustering algorithm (Scott and Knott, 1974). "Erroneous" pixels (=having NaNs as resulting maxima) were excluded from the cluster process in order to avoid disturbances. The data was rescaled to make the characteristics more comparable with each other. If the number of clusters is not predefined, it can be identified by drawing an elbow chart. For different numbers of clusters K the measure of the variability of the observations within each cluster (Total within-cluster sum of squares, y-axis) is calculated and the curve should bend like an elbow at the optimal value. Since the algorithm did not suggest a number of clusters, we chose six clusters for a sufficiently detailed analysis since it gave consistent results when repeating the automatic algorithm for several times (each time the algorithm clusters slightly differently)." [end of section]

- Furthermore, I do not understand the concept behind Figure 8b. It shows the maximum difference before and after a gap, but what does this mean if >50% data are missing as e.g. in the northernmost part in Germany? Could you explain this more clearly?

**Response:** Thank you for your comment. Are you referring to Figure 2b instead of Figure 8b? The reason for >50% of missing data is because radar coverage started relatively late for the concerning pixels (sometimes only from 2014 onwards or so), thus the time series are a lot shorter. The Figure should support that NaN imputation might be problematic for our data. Other reasons than the large amount of data is that firstly, too many pixels are with quite a lot of missing values (4% of data means already almost 70,000 timesteps of 5 minutes). Furthermore, the imputation bridge may yield very high values adding large uncertainty. As mentioned in the text, imputing unrealistic high values in these gaps is likely to add bias to our results more severely than if we just keep everything as is. Maybe the NaN section might be unnecessary for you, we however feel that it is more transparent to display our thoughts behind what we did.

- Why are the values generally higher the closer they are to the radar site? Could this also point to the different behaviour of extremes depending on the distance from the radar?

**Response:** The values are lower in the outer areas of the radar circles potentially due to the overlap of radar information in these areas. Fewer and shorter NaNs sequences in the time series will also reduce the number of high imputation values.

**Minor comments**

- In general, the manuscript should be proof-read again, there a many awkward formulations,spelling mistakes, etc.

**Response:** Sorry for that! We have sent the revised document to proof-read again.

**Specific Comments:**

- P1L17ff.: This whole sentence sounds weird, fatal disasters don't react to anything

**Response:** see above (first paragraph in new introduction)

- P1L20: Introduce the acronym PMP here

**Response:** We added it.

- P2L24: PMP can be estimated

**Response:** We changed it.

- P2L52: AR(1) -> first-order autoregressive process?

**Response:** We changed the sentence to "Zhang et al. (2013) showed that the scaling exponent varies around 0.5, if the vertical moisture flux and rainfall can be modelled as a censored (or truncated) first-order autoregressive process AR(1)."

- P3L58: Breña-Naranjo (this needs to be corrected in the references as well)

**Response:** Sorry! We changed it.

- P3L63: 16 years

**Response:** We changed it.

- P4L77: Aren't there currently 17 C-Band Radars?

**Response:** Sorry! We change it.

- P4L78: delete "free and purchasable"

**Response:** We deleted it.

- P4L81: ground information

**Response:** We changed it.

- P4L90f.: I find the justification that "Due to comparison reasons with another study at our institute only years 2001 to 2016 had been used for this study" rather weak. It would have been worthwhile to use the data until 2018, since you also mention that "With longer available time series of radar in the future, the deviation can be further investigated and tested" in the discussion.

**Response:** Thank you for your comment, we are sorry that we cannot find a new reason to make the justification sound better. We agree, that longer time series are always better, however, we would need to find a better data handling approach in order to analyse years 2017 + 2018. The preparation of the manuscript together with the final analyses took longer than expected. With the current data handling approach it would take too long to 1) process the data, 2) redo the complete analysis and 3) interpret the results, that is why we kept the analysis to the years 2001 – 2016. We will see in further work if our argumentation was correct or not. Thank you for your understanding!

- P4L95-100: there is no need to mention the data size or how the data was saved.

**Response:** The data size seems very relevant for us. Without high-performance computing and sufficient storage capacity, it will not possible to do complex analysis of RADKLIM-YW in a reasonable

amount of time. Due to these reasons, it took a while to convert to time series (obviously better programmers will have no problems with this issue).

We removed the following passage: "Analysis was conducted in R ......... was chosen to store the data." (2 sentences)

- P4L99: Why don't you use "NaN" for missing values?

**Response:** We changed NA to NaN in all 6 occurrences.

- P5L103-105: Are data of overlapping radar coverage areas similar? Since the data was measured by two different radars, the values can differ significantly (e.g. Yan and Bárdossy, 2019)

**Response:** We totally agree that radar data within overlapping radar cones can differ depending on the radar. In lines 103-105 we are however not evaluating the quality of the radar data, but focus on the data coverage, which increases when having more available (even different) information. The work of the DWD has been to merge the different radar information in a consistent way.

- P5L107ff.: This whole paragraph is difficult to read, please rewrite this in a clearer way.

**Response:** We edited the text as follows:

(P5L107ff):

"It is hard to handle NaNs in highly episodic geophysical events such as rainfall. Based on Fig. 2, we chose to not do any data interpolation, since the consequence of imputing potentially too high extreme values is more severe and uncertain for our study than the missing of any extreme values."

- P6L115: How was the aggregation done considering the missing values? And how where events separated? Did you use a threshold? If yes, which?
  P6L116: Durations of up to 3 h or 3 d? This whole sentence is difficult to read.
  P6L130: Is the scaling relationship formulas are not correct

**Response:** The aggregation was done with rolling sums applied over moving windows (compare text edit below), ignoring the missing value (=treating them as zero when "rolling over" them). The evaluation is not event-based but time based, thus events are not separated. We also have done the analysis event-based out of interest, but obviously most events stop after a few days, thus this approach is not useful if looking at maxima across different scales. The event-based analysis will also not necessarily give the maxima for a certain time period, since some maxima are the sum of several short heavy events.

The authors agree that the methodology part might create unnecessary confusion (following Referee #1's comment). Thus, we merged sections 2.2.1 and 2.2.2 into one paragraph 2.2 "Depth-Duration relationships" and removed most of the equations for a better reading.

The methodology section is now shortened as follows:

"2.2 Depth-Duration relationships

Maximum rainfall values for each duration τ between 2001-2016 were calculated with rolling sums applied over moving windows using the R package RcppRoll (Ushey, 2018). Durations of up to 3 d were chosen for the analysis, with multiple steps for minutes and hours out of our interest for sub–hourly and sub–daily pattern. The records may include non-rainfall data and thus do not imply continuous precipitation for the period considered. Values were not aggregated spatially, since this usually reduces the maximum intensity values (Cristiano et al., 2018). First, the extreme values for each pixel and duration $M_{max}^{\tau,pixel}$ are calculated. Afterwards, the overall maxima for whole Germany for each τ ($M_{max}^{\tau}$) is extracted from these calculated extreme values. Based on these results, the depth–duration relationships can be built for each pixel as well as for the whole of Germany."

- P7L138: Please reformulate sentence (also see major comments above)

**Response:** Please see our text edits that are included at the response to the corresponding major comment above.

- P7L143: The temporal resolution of the ground truth reference should be mentioned. Furthermore, "world record" should be used (also in caption of Fig. 4)

**Response:** We used "world record". We have tried to find the temporal resolution on which the world record curve is built. However, even when checking papers that individually treat certain historical rainfall events/maxima, the temporal resolution was not found for most cases.

- P8L149: what are "very distant places of Germany"? Distant from what?

**Response:** This means that the places are not close to each other, rather distant from each other. We edited the sentence as such: "..even though the maxima are observed rather randomly across the whole of Germany."

- P8L151f.: Which temporal resolutions did you use for your analysis? 5 Min increments up to 16h? 3 days? Please specify!

**Response:** All analysis is based on the 5 min temporal resolution of RAKLIM-YW. P6L116: Maximum rainfall intensities were retrieved from aggregating the 5 min values for each duration of interest …..

- P8L156: Data uncertainty is mentioned here but it's effect is not discussed!

**Response:** Data uncertainty is also mentioned in the data description; however, the effect cannot be estimated well, without doing a separate study on the radar QPE data processing scheme/algorithm itself. This is partly already done by the DWD, for example in Kreklow et al. (2019), however, we expect the DWD to do further publications about it. We could find many references discussing the effect of radar rainfall uncertainty on hydrologic responses, but not on scaling behaviour. We guess this is because this is one of the first studies in this regard. If the reviewer suggests references, we will carefully address them in the manuscript.

- P8L163: the first two quantiles are identical

**Response:** We changed the first one to 0.99999

- P8L167: development the rainfall-duration ! development of the rainfall-duration

**Response:** Thank you! We changed it.

- P8L170f.: The statement that extreme rainfall events share common characteristics such as peak rainfall depth and correlation structure regardless of time-scale is a 'strong' statement that somehow contradicts the fact the rainfall extreme are spatially and temporally variant and their correlation structure differs.

**Response:** We agree to the referee and have removed the statement.

The final sentence of Chapter 3.2 is changed to: "Contrary to this conjecture, the curves in Fig. 5 (99.9% and 99 %) show a rather smooth scaling behaviour."

- Figure 6: A discrete colour bar should be used, moreover the spacing between the durations does not reflect the real spacing. An additional suggestion would be to add a second colour bar showing the associated rainfall values. This can help relate the quantiles to the rainfall values.

**Response:** We changed the colour bar to a discrete one (see Figure below), but think that a second colour bar will add more confusion than that it helps. We further believe that a reader can relate the quantiles to the rainfall values with the help of Figure 5 if necessary.

[Figure]

FIG. 2: Figure 6 with new color bar. Caption: Locations of the 0.99999, 0.9999, 0.999, and 0.99 quantile rainfall with varying durations from 5 min to 3 d. Point colors represent the corresponding rainfall duration, similar for each quantile. Different numbers of data points in panels a-d result from several data points being at the same location.

- Figure 6: Why are the 0.99 quantiles are mostly located in the south of Bayern?

**Response:** We have edited the text corresponding to Figure 6 (P8L163 ff) and hope that it will answer your question:

"It shows that the number of locations increases the lower the quantile of maximum rainfall is. This suggests the reduction of the influence of one single rainfall event on the depth-duration relationship causing inflection in the curve. Additionally, from a certain degree of quantile (Fig. 6 d) the locations of maximum rainfall contributing to the development of the rainfall-duration relationship seem to happen mainly in the wider Alpine region in South Germany. This suggests that rather natural rainfall mechanisms are dominating the scaling relationship, such as regional characteristics and meteorological conditions (e.g. orographic lifting or leewards effects)."

- Figure 7: please redo with a discrete colour bar and maybe scale it to the values so the details in the map are more visible.

**Response:** If the referee would find it more suitable, we could change the colour scheme as given in the following example (colours represent quantiles):

[Figure]

- P12L180: avoid formulations such as "really high"

**Response:** We changed it to "These maxima seem to be dominated by single events or single heavy rainfall occurrence."

- P12L185: Stuttgart

**Response:** We changed it.

- P12L187: who

**Response:** Sorry! We changed it.

- P12L188 what do you mean with "real rainfall process"?

**Response:** "Real" corresponds to "observed" in comparison to "modelled/simulated" process. We edited the sentence to: "This result also implies that natural rainfall processes significantly deviate from the assumption of the simple rainfall models suggested by ...... "

- Figure 8: How are curves for the cities calculated, mean of all cells in the city or maximum cell? This might be relevant to explain why neighbouring cities show very distinct behaviour.

**Response:** We changed the caption of Figure 8 as follows: "Depth-duration relationships of rain records for single pixels at rain gauge locations within state capitals of German federal states."

The gauge locations as x-y coordinates (provided by the DWD via an online repository) are transformed into RADKLIM projection (with potential slight geographic shift due to assignment of point data to raster data). We chose these cities since they are relatively well spread over Germany. We could have also selected random pixels, but preferred to choose known locations.

We don't understand the second part of the comment. The minimum distance between the cities is approx. 80 km, a distance which usually guarantees different characteristics. Even in the small structure of Germany, already the two different oceans influence the meteorological characteristics significantly besides multiple other influences (continentality, luv/lee-ward effect, ..). Only two cities, Mainz and Wiesbaden are rather close to each other. What distinct behaviour of the two needs explanation?

- P13L193ff. Rewrite this paragraph and be more specific, what is "successfully classified", what is a "certain colour", etc.

- P13L193: dept-duration ! depth-duration

**Response:** We have edited the paragraph as follows:

"The maximum depth-duration relationships for all pixels within Germany were clustered since Fig. 8 indicated that they might show similar shapes. The k-mean clustering algorithm classified the depth-duration relationship into six categories revealing different curve characteristics regarding the curve shapes. Figure 9 shows a categorical map of Germany representing each category with an individual colour. Additionally, depth- duration relationships at 100 randomly chosen grid elements from each category are shown with the regression line from Category 5 as reference."

- P13L199: "that pour for around 1 h and move on or weaken" -> rewrite this

**Response:** We have edited it as follows:

"The behaviour of the curve between 5 min and 1 h is associated with strong convective rainfall events of around 1 h within the corresponding pixel."

- Figure 9: Does this relate to topography?

**Response:** We would partly relate it to topography as orographic rainfall does play an important role for certain rainfall durations. As written on P15L209 ff, we attribute some large clusters to orographic rainfall, whereas some clusters can be identified as other large scale events without a direct relation to topography.

- Figure 9: Legend of plot 'mm/uration' ! mm/duration

**Response:** Following the first referee's recommendation, we have changed all mm/duration to simply mm.

- Figure 9: How would the clustering and this map look, if the data was divided between summer and winter period? Did you look into this?

**Response:** We unfortunately did not look into a division between summer and winter. It is a good idea for a follow up analysis!

- P15L204f.: If the look similar and occur together why do you distinguish these categories? (c.f. major comments above)

**Response:** Please compare our answer to the corresponding major comment above. We preferred to first to the automatic grouping, afterwards reduced the groups as best as possible with our expert knowledge.

- P15L205: a slope is steeper instead if higher

**Response:** Thank you! We have changed it!

- P15L213: the term 'super-daily' is confusing, please consider changing it.

**Response:** We believe that the term is an appropriate way of addressing everything beyond "day".

- P15L219: saying that areas with category 5 have never been hit by any 'extreme' extreme event needs more evidence. It could be that the occurred events were not well captured due to data uncertainty.

**Response:** Thank you for your comment! We made this statement based on what we get from the data. Every pixel's "extreme" extreme value for different duration was extracted, thus it is not wrong to say that the pixels of category 5 show certain characteristics. In our opinion, we generally have no "hard" evidence" that products from remotely sensed data deliver "true" information, especially for remote areas with lack of supporting information, we mention the data uncertainty in the beginning and should then work with what we got.

We added in the beginning of the corresponding sentence in L219: "Based on the data set, these regions/locations ... " and hope that it makes it clear enough.

- P15L232: areas don't "experience" a rainstorm...

**Response:** We changed it to:

"… at a given point varies location by location based on the occurred rainstorms." [removing the "areas"]

- P16L232: … the same goes for pixels!

**Response:** We changed the sentence in L262:

"The shape of the curve was governed by the temporal structure of the extreme rainfall events at the pixel location."

- P16L264f.: Reformulate this sentence

**Response:** We edited it as follows:

"The scaling behaviour thus can be significantly different for each pixel, because the rainfall characteristics for each pixel are very different as well."

- Figure 10: Please add legend and increase the grid visibility.

**Response:** We added a text in the caption and increased the grid visibility

Caption 10: Dependency of maximum depth-duration relationship characteristic on underlying pixel sample size. The maximum rainfall values are derived from (a) 10, (b) 100, (c) 1000, and (d) 10000 random pixels from all considered pixels (n=392 128) within Germany. For each sample size, 30 ensembles are displayed and compared to the overall maximum curve from Fig. 4 and 5 (yellow top line in (a) – (d)).

- P17L268f. If you have the data until 2018, why didn't you use them? (c.f. P4L90f.)

**Response:** compare answer to P4L90f.

References: Several issues with capitalization of titles and author's names.

**Response:** Thank you for your helpful comments. The data was mainly extracted via the doi automatically in JabRef/Mendeley. Afterwards, we have revised the references to the best of our knowledge. We don't know exactly which titles you are referring to, but we think that we took the capitalization as given in the journals, e.g., Jennings 1950 and Paulhus 1965: all capitalized (Monthly Weather Review had it that way at the time). "Intensity-Duration-Frequency" is sometimes capitalized, sometimes not. We kept it in all cases as in the journal.

We have corrected the authors' names in:
- Breña-Naranjo et al. 2015

We have changed the following references after revisions:
- We have removed Marra et al. 2016 since it is the preprint of Marra et al. 2017
- We have replaced Cristiano et al. 2018 (preprint) by the final revised paper
- We have updated Lengfeld et al. 2019 (preprint) to the final revised paper

We also have removed some double urls.

**Reference**

J. Kreklow, B. Tetzlaff, G. Kuhnt, and B. Burkhard. A Rainfall Data Intercomparison Dataset of RADKLIM, RADOLAN, and Rain Gauge Data for Germany, Data, 4, 118, https://doi.org/10.3390/data4030118, 2019.

---

## Author Response (AR2)

*nhess-2020-192 (Pöschmann et al.): Author's Response to Handling Editor*

Dear Professor Pinto,

we greatly appreciate the very constructive comments from Referee #2 that helped to clarify remaining problems with the manuscript. We have provided replies to all comments and questions and mostly agreed to change the manuscript as suggested.

The revised manuscript was finally carefully proofread by a native English speaker at our institute. The revision has been approved by all co-authors.

Thank you for your consideration!

Sincerely,

Judith Pöschmann on behalf of all authors

**nhess-2020-192: An analysis on temporal scaling behaviour of extreme rainfall of Germany based on radar precipitation QPE data (Pöschmann et al.)**

**Reply to Referee #2**

We thank anonymous referee #2 for his/her detailed comments and suggestions in order to improve our revised draft. We have provided responses to all comments in blue and have updated the manuscript accordingly. All line numbers given in our responses refer to the "old" manuscript with track changes.

**Report:**

Due to the limited time for this review round, I could only take a look at the revisions of my comments (and not the ones from all reviewers). Since the paper has been modified substantially, there is the possibility that my comments may contradict the ones from the other reviewers.

Most of my major concerns from the first round of revisions have been addressed. There are however still two issues that remain

**Main Issues**

- Clustering: I acknowledge your line of argument that the actual number of clusters is not the main focus when making a point that the scaling can be classified based in the inflections points. And maybe you can justify 6 clusters, but when looking a figure 8 (in the revised manuscript), categories 1 and 2 or 4 and 5 look very similar to me. So why not go with 4 categories instead? The interpretation of the results in section 3.5 is based on those 6 categories but this interpretation may turn out quite different when fewer categories are chosen (more probably won't make any sense). And if a different number of clusters leads to different results and subsequently to a different interpretation that would an issue that needs to be addressed. If not, then it would support the choice of six classes.

Did you look into this and could you comment on this? Maybe a supplement or appendix showing results for different clusters would be an option?

**Response:** We think it is a good suggestion and added different cluster results as supplement for k = 3, 4, 5 and 7. Find below the maps of the corresponding cluster size as well as the table. As you can see the interpretation of the clustering does not stop making sense when looking at different clusters. The higher the number the more distinguishable results would be possible, but we stopped at a cluster number of six, because it made most sense for us.

[Figure]

[Figure]

I still don't understand why you are going into the imputation bridge part. You are definitely correct that an interpolation of NaN does not make any sense in the case of rainfall. But you could justify this simply by stating that there could be e.g. extremes that occurred during a gap (and thus remain undetected) or that an interpolation between two values before and after a gap would lead to false information because you do not know what happened in between. But what is the message you a trying to convey in Figure 2b? A map with maximum values between gaps does not really make sense to me because it is not stated how many gaps there are in total and how long these gaps are. Some statistics about the number and length of these gaps would be more meaningful from my point of view.

**Response:** We accept to not go into the imputation bridge part if it raises more questions than it helps our argumentation line. We have removed the related sentences from the manuscript and also Figure 2b

**Minor comments (line numbers referring to the manuscript with track changes)**

First of all, the manuscript should be proofread again. There are still numerous errors, especially in the changed passages. Furthermore, for the sake of readability, some sentences should be formulated in a more direct and compact manner.

**Response:** We have given the manuscript again to a native English speaker at our department and he has carefully proofread the document.

In general:

- avoid e.g. "spatio-temporal" vs. "spatio temporal"

**Response:** We have consistently changed everything to "spatiotemporal"

- try to avoid double brackets like "(Fig. X (a))" or similar

**Response:** We have removed all double brackets.

- on more than one occasion you write that "pixels experience" something. This sound strange and I'm not sure if this is correct English. The same goes for e.g. "the entire of Germany", "the whole of Germany" and similar.

**Response:** We changed everything to "all Germany" when referring to Germany as a whole.

Numbers: use "," for separating thousands, i.e. 1,000 etc. This is inconsistent throughout the manuscript, including the figures!

**Response:** We have carefully revised this and think it is consistent now through the manuscript, including the figures.

Regarding the use of "world record": I made a mistake in my first review, I wanted to say that this term should NOT be used, sorry about this. In my opinion it sounds more like yellow press than scientific paper. Maybe something like "global precipitation extremes" is better...

**Response:** We changed it

**Specific corrections (without claim for completeness)**

Abstract: I suggest you write years, hour and day instead of yrs,h, and d here

**Response:** We agree and changed it.

- l. 1 investigated (the rest is past tense as well)

**Response:** Thank you! We have changed it.

- l. 19 "urban and non-urban flash floods" → are there any others? Just say "flash floods"

**Response:** We agree and changed it.

- full stop missing the same line

**Response:** You are right, it looks like it is missing in the revised version, potentially latexdiff did not convert it correctly. It is not missing in the new document.

- l. 21 "needing"? Sounds strange, rephrase

**Response:** We changed it to "requiring".

- l. 22. "Obstacles to identifying and investigating extremes" → sounds strange, rephrase.

**Response:** We have changed it to: "It is difficult to identify and investigate extremes and record rainfall events because of their rare occurrence as well as ….."

- l. 23ff. makes no sense, "50% obwhich observed extremes"(?), "even more"(?) This is not clear, be more specific!

**Response:** For the sake of clarity we changed the sentence to: "Lengfeld et al (2020) analysed the problems of rain gauge observations, that miss more than 50% of the extreme rainfall events observed, especially with data of higher temporal resolutions."

- l. 30 from may point of view, attenuation is also a major reason for uncertainty because this may reduce extremes

**Response:** We have added it to the list.

- l. 34 this is a 1:1 quote from the AMS Glossary of Meteorology. Mark this accordingly (or reformulate) and update the source to American Meteorological Society, 2020: Climatology. Glossary of Meteorology, http://glossary.ametsoc.org/wiki/

**Response:** We have marked it accordingly and updated the source.

Figure 1: I suppose you removed the triangles because another reviewer suggested this. I my opinion, the first version with the triangles was better….

**Response:** We also like the version with triangles, however, one reviewer suggested to make it consistent. As we do not have "single values" for the Regional extremes of Germany, we chose to only take the regression lines from all three examples shown.

l. 82 change to "extremes found"

**Response:** Thank you! We have changed it.

l. 85 "…a reprocessed…" delete "carefully"

**Response:** We deleted it.

l. 101f. change to "One QPE from German radar data RADOLAN (German: RADar OnLine Aneichung)" and add a citation

**Response:** You are right, we have added the following reference:

Winterrath, T.; Rosenow, W.; Weigl, E (2012). On the DWD Quantitative Precipitation Analysis and NowcastingSystem for Real-Time Application in German Flood Risk Management. Weather Radar and Hydrology, Proceedings of a symposium held in Exeter, UK, April 2011. IAHS Publ.,351. 323–329.

l. 111 "two formats" You mean "two versions", I suppose the format is the same

**Response:** We changed it to "two versions".

l.117 due → because of

**Response:** Thank you, we changed it.

l. 118 "have been"

**Response:** Thank you, we changed it.

l. 123: "values (compared below) compared to RADOLAN" repetition, rephrase

**Response:** We are sorry for the confusion here. The bracket "compared" refers to the general passage about missing values in RADKLIM, the second "compared" refers to the comparison with RADOLAN. We have removed the two words in brackets and have shortened the sentence a little.

l.125 I still think that the details about files sizes are not really relevant and this sentence can be deleted. If you decide to keep it nevertheless: "making" sounds wrong in this context

**Response:** We agree and shortened the passage to:

"The data is available as one layer for each time step. Since not all raster pixels are ....."

l. 131 "Some time series" → you mean "data in some areas"?

**Response:** We have changed it.

l. 136 "...red spots could mean..." sounds strange, rephrase

**Response:** We have removed Figure 2b and its related sentences.

Caption Figure 2 "needs"

**Response:** We changed it.

l. 144 "than the missing of any extreme values." sounds strange

**Response:** We simplified it to: "than missing extreme values".

l. 149ff difficult to understand, reformulate

**Response:** We changed the sentence to: "Time windows of up to 3 days were chosen for the analysis, with special focus on the sub-hourly and sub-daily durations."

l. 164 "built" → developed?

**Response:** We agree and changed it.

l. 171 it is called "k-means", sometimes the K is also capitalized, please check and use this consistently

**Response:** This is obviously true, thank you for notifying us. We have changed it accordingly, using K capitalized.

l. 198: "xx minutes" (?)

**Response:** Sorry if this was not clear! We have deleted the brackets and its content.

l. 199f "Hesse state" vs. "Bavaria" → stick to one convention. Furthermore, what the purpose of mentioning that this has not been documented in the news?

**Response:** We removed the "state" from Hesse and removed the passage about the non-documentation in the news.

l. 205 What are "maximum locations"? Do you mean the locations of maxima?

**Response:** Yes, we do and changed it.

Table 1 is a good idea, but much too detailed. Reduce to lat/lon to 2 trailing digits, precipitation to max 1, use CET and 24h Format for times, Location is WGS84 not WG84

**Response:** We provided the details since one referee asked as to do it for being able to check the results. We edited the table according to your suggestions: Reduction of lat/lon to 2 trailing digits, reduction of precipitation to max 1, used CET and 24h format for times and changed WG84 to WGS84

l. 209ff consider deleting brackets with the ranks, is does not enhance the readability. It's sufficient to mention this in Figure 4

**Response:** We agree and deleted it.

l. 213 "3-regime-form" ->?

**Response:** We changed it to "three phase regime" as used in line 210.

l. 215 "Figure 5 shows the location of the high quantile rainfall" → delete the numbers for the sake of readability

**Response:** We agree and deleted them.

l. 216 reformulate this sentence, this is not correct

**Response:** We are sorry, that this sentences caused a problem and for the sake of better understanding, we rephrased it as follows:

Old version: "It shows that the number of locations increase the lower the maximum rainfall quantile is."

Update "It shows that at the highest considered quantile (0.99999) multiple maxima appear at similar locations, potentially referring to the same rainfall events, whereas for lower quantiles (e.g., 0.9999 to 0.99), maxima are more spread over Germany and the visible points increase in number."

l. 218-220 same as l. 216, hard to understand, rephrase

**Response:** We simplified the sentence as follows:

Old version: "Additionally, from a certain degree of quantile (Fig. 5 (d)) the locations of maximum rainfall contributing to the development of the rainfall-duration relationship seem to happen mainly in the wider Alpine region in South Germany.

New Version: "Additionally, locations of such high quantile maxima (e.g., 0.99 quantile in Fig. 5) seem to occur predominantly in the wider Alpine region in South Germany."

Legend figure 4: "392,128 in total" or "n=392,128"

**Response:** We have changed it to "n=392,128".

l. 230 please note that the red and yellow spots are difficult to see in a printed version

**Response:** We agree and tried several colour scales before to avoid that version. The authors' compromise was the current colour scheme.

l. 233 similarly located → rephrase

**Response:** We changed it to: located at similar places in the maps.....

l. 241 why don't you mention that the reference is a single power law right here?

**Response:** We agree and changed it according to your suggestions.

l. 245 what does "attached to the maximum rainfall event" mean? Rephrase!

**Response:** Sorry, if "attached" was not understandable. We have changed it to: ".... small (or zero) rainfall observations around a maximum rainfall event." We think that this clarifies that Galmarini et al. (2004) are talking about values before and after very high rainfall sequences.

l. 254 "The k–mean clustering algorithm successfully classified the depth–duration relationship into six categories" -> Of course, because you defined 6 classes, what's successful about this?

**Response:** If we are talking about the manuscript with tracked changes, this word has already been removed!

l. 257 "grid points" not elements

**Response:** We changed it.

l. 257ff. Why to you capitalize "Category"?

**Response:** We thought that the capitalizing enables a better reading. However, we removed the capitalization.

l. 261 "beginning of the curve"

**Response:** We changed it.

l. 265 smaller pixels → I assume that your pixel size is constant

**Response:** True! We changed it to: Smaller areas.

l. 267 "Category 1's" possessive s in combination with a number looks strange

**Response:** We changed it to "the slope of category 1".

l. 268 repetition of "location", rephrase!

**Response:** We have removed the comparison part of the sentence.

Caption Fig. 8: Rephrase to "Resulting clusters of the maximum depth-duration relationships"

**Response:** We changed it.

Figure 9 "??" in caption, missing reference?

**Response:** Sorry, we updated the reference.